# One-Round Active Learning through Data Utility Learning and Proxy Models

**Jiachen T. Wang**⋆  *tianhaowang@princeton.edu*
*Princeton University*

**Si Chen**  *chensi@vt.edu*
*Virginia Tech*

**Ruoxi Jia**⋆  *ruoxijia@vt.edu*
*Virginia Tech*

**Reviewed on OpenReview:** *https://openreview.net/forum?id=8HQCOMRa7g*

## Abstract

While active learning (AL) techniques have demonstrated the potential to produce high-performance models with fewer labeled data, their application remains limited due to the necessity for multiple rounds of interaction with annotators. This paper studies the problem of one-round AL, which aims at selecting a subset of unlabeled points and querying their labels *all at once*. A fundamental challenge is how to measure the utility of different choices of labeling queries for learning a target model. Our key idea is to learn such a utility metric from a small initial labeled set. We demonstrate that our approach leads to state-of-the-art performance on various AL benchmarks and is more robust to the lack of initial labeled data.

In addition to algorithmic development and evaluation, we introduce a novel metric for quantifying '*utility transferability*' – the degree of correlation between the performance changes of two learning algorithms due to variations in training data selection. Previous studies have often observed a notable utility transferability between models, even those with differing complexities. Such transferability enabled our approach, as well as other techniques such as coresets, hyperparameter tuning, and data valuation, to scale up to more sophisticated target models by substituting them with smaller proxy models. Nevertheless, utility transferability has not yet been rigorously defined within a formal mathematical framework, a gap that our work addresses innovatively. We further propose two Monte Carlo-based methods for efficiently comparing utility transferability for different proxy models, thereby facilitating a more informed selection of proxy models.

## 1 Introduction

While the past several decades have witnessed the potential of active learning (AL) methods to produce high-performance models with fewer labels, they are often slow and even infeasible in practice. Specifically, for existing AL methods to be effective, they require many rounds of interaction with the annotators and the interaction in different rounds must be adaptive, in the sense that the choice of the instances in one round depends on the responses to the labeling requests in all the previous rounds. However, for tasks in which it takes significant time and resources to label the queried instance (e.g., in scientific experimental design, it would take days or even months to obtain feedback from wet-lab or physics experiments (Botu & Ramprasad, 2015; Nord et al., 2016; Yang et al., 2019)), adaptive interaction over many rounds is inefficient. Moreover, in practice, there is typically a waiting time before labeling queries are responded to, simply due to the fact that potential data annotators may not respond promptly upon request. For example, in marketing applications, one can actively request feedback by sending surveys to customers (annotators), but the customers may not

---

⋆Correspondence to **Jiachen T. Wang** and **Ruoxi Jia**. Jiachen T. Wang did the work at Harvard University.

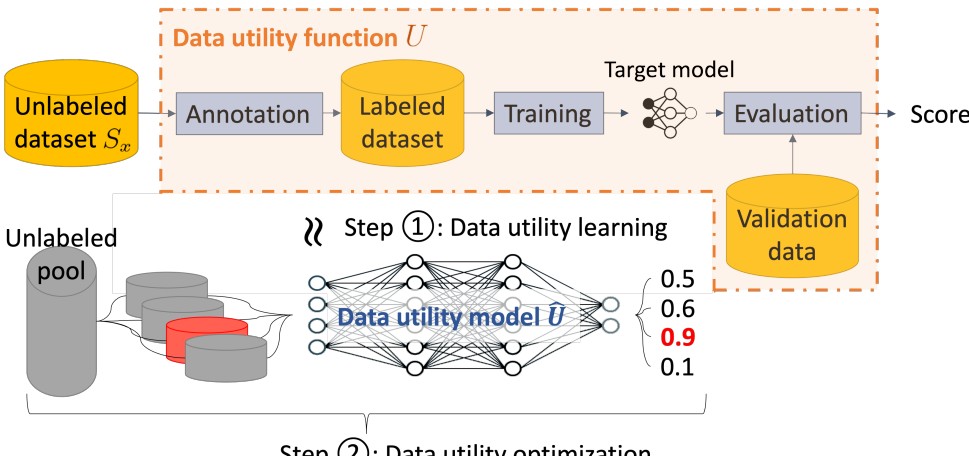

Figure 1: Overview of DULO.

be responsive (Sharp et al., 2011). If the overall task is to be completed within a certain time frame, this effectively limits the number of rounds of interaction.

This paper focuses on one-round AL—a special AL problem where only a single round of interaction is allowed. Specifically, the learner is required to select a set of unlabeled instances and query their labels *all at once*. Importantly, this setting features high throughput and time efficiency as the selected instances can be labeled completely in parallel. A fundamental question in one-round AL is: *How can we measure the utility of different subsets of unlabeled data points and select the one with the highest utility while adhering to some given labeling budget?*

A natural idea to approach the one-round AL problem is to adopt data utility metrics from state-of-the-art multi-round AL methods (Wei et al., 2015; Ash et al., 2019; Killamsetty et al., 2020) and use them to perform one-round selection. A commonality of most of these data utility metrics is that they rely on *hypothesized labels* assigned by the temporary target model that is trained on the currently available labeled instances. For multi-round AL, the size of labeled instances will grow as more rounds of labeling requests are responded to by the annotators; as a result, the hypothesized labels will become more accurate over time and the data utility metrics based on hypothesized labels will, in turn, become more reliable. However, in one-round setting, the classifier used for generating hypothesized labels can only be trained on a small initial labeled set and thus the hypothesized labels are highly unreliable. Hence, directly adapting multi-round AL methods to one-round setting suffers unstable and unsatisfactory performance. There are only very few existing works in the field of AL focusing on the one-round setting. Most of them, however, either make too strong or too weak assumptions about the initial labeled data points. For example, Contardo et al. (2017) assumes a large-size labeled pool comprising classes that do not appear in the unlabeled pool, and Yu et al. (2006); Gu et al. (2012); Shoham & Avron (2020); Jin et al. (2022) consider a very restricted setting where *no* initial labeled data points are given. On the contrary, this work considers a more reasonable setting where there are *few* initial labeled data points given.

The contributions of this work are summarized as follows.

**(1) New algorithm for one-round AL.** We propose **D**ata **U**tility function **L**earning and **O**ptimization (DULO), a model-agnostic one-round AL framework (illustrated by Figure 1). Our framework is grounded on the notion of a *data utility function*, which maps any given set of unlabeled instances to some performance measure of the target model trained on the set after being labeled. One-round AL can be naturally formulated as a problem of seeking the set of unlabeled instances that maximizes the data utility function. We propose to *learn* a model to approximate the data utility function, and present a blockwise stochastic greedy algorithm to efficiently select data that optimizes the learned model. As this model is trained to predict data utility without label information, thus circumventing the limitation caused by unreliable hypothesized labels. To improve scalability, we leverage the *proxy model technique*[1] which uses a smaller proxy model in place of the larger target model to perform data utility learning. To further address the concern about the computational

---

[1]The proxy "model" here means the overarching learning algorithm that is used for accelerating active learning.

efficiency of data utility learning, in Section 3.2.1 we show that it is not necessary to learn the utility of large data subsets as the variance of utility over different subsets of a given size diminishes as the size increases.

**(2) Characterizing and measuring utility transferability for selecting proxy models.** The proxy model-based idea above builds upon a popular observation that the choice of training data often affects the performance of different models in similar ways. Specifically, the data that are more useful for training one model are usually also more useful for training another, even when the two models are of different complexities. Indeed, this observation has been leveraged to improve efficiency in various applications, including AL, coreset selection, data valuation, among others (Lewis & Catlett, 1994; Coleman et al., 2019; Nath et al., 2021; Jia et al., 2019; Wang et al., 2023). Despite the broad applicability, principled ways of measuring the correlation of data utility between different models are still lacking. Towards this end, we formalize the notion of *utility transferability*, which characterizes to what extent a proxy model mimics the target model's performance variations when trained over different subsets. We further develop two Monte Carlo methods to efficiently compare the strength of transferability for different proxy models, which could facilitate the selection of a better proxy model.

**(3) Experiments.** We evaluate DULO to select data from a class-imbalanced pool, a noise-contaminated pool, and a unlabeled pool that exhibits natural variations of data quality. We show that our approach outperforms state-of-art batch AL strategies across different settings and datasets. In addition, it is more robust to the lack of initial labeled data.

## 2    Related Work

According to whether the samples are selected iteratively or not, AL approaches can be classified into two main categories: One-Round Active Learning and Multi-Round Active Learning.

**Multi-Round AL.** Multi-round active learning has been widely studied in the past, where one can send selected unlabeled data to be labeled to the oracle, update the querying strategy (usually a partially trained model), and choose new unlabeled data depending on the updated querying strategy. We refer the readers to (Ren et al., 2021) for a more comprehensive survey of multi-round AL. Earlier studies of multi-round AL focused on the setting where only one example is selected in each round, e.g., (Schohn & Cohn, 2000; Fine et al., 2002). Batch AL was later proposed to improve label acquisition efficiency by querying multiple examples in each round. Most of state-of-the-art multi-round AL methods rely on partially-trained classifiers to generate *hypothesized labels* for unlabeled points and further use them to prioritize them for selection. For instance, BADGE (Ash et al., 2019) samples a batch of unlabeled points with large and diverse gradients, but to calculate the gradients, unlabeled data need to be assigned with some hypothesized labels. Filtered Active Submodular Selection (FASS) (Wei et al., 2015) selects unlabeled data points by optimizing the predicted performance of a K-Nearest-Neighbors classifier or Naive Bayes, which can be calculated analytically only given the hypothesized labels of the unlabeled points. GLISTER (Killamsetty et al., 2020) formulates the active learning problem as bi-level optimization, which seeks unlabeled points that lead to the highest validation performance when these points and corresponding hypothesized labels are used for training. These approaches suffer unsatisfactory performance in the early rounds as the classifier used for generating hypothesized labels is trained on limited labeled data. In contrast, our data utility metric estimates the unlabeled data's usefulness without the need of hypothesized labels.

**One-Round AL.** One-round active learning is a much less studied field compared with the multi-round setting, as it cannot benefit from the feedback of the labeling oracle. However, such techniques are useful when frequently querying the labeling oracle is not possible or time-consuming, e.g., as cited by (Gu et al., 2012; Contardo et al., 2017) when using Amazon Mechanical Turk (AMT). An existing one-round AL approach is to derive an upper bound on the prediction error of the target models and then select unlabeled data by minimizing the bound. However, such a derivation is possible only for specific models, such as linear regression (Yu et al., 2006; Gu et al., 2012) and graph-based predictive model (Guillory & Bilmes, 2009). By contrast, our work proposes a general-purpose approach for one-round AL and does not restrict the type of target models. Another idea of enabling one-round selection is via meta-learning (Contardo et al., 2017). However, unlike our work, this work assumes a large-size labeled pool comprising classes that do not appear in the unlabeled pool. Shoham & Avron (2020) selects unlabeled data in one-round via optimizing a V-optimality criterion for kernel ridge regression. Jin et al. (2022) proposes a one-round AL technique for

image segmentation tasks based on contrastive learning and diversity-based querying strategy. Compared to our work, Yu et al. (2006); Gu et al. (2012); Shoham & Avron (2020); Jin et al. (2022) consider a slightly different setting where *no* initial labeled data points are given and their algorithms are specialized to specific target models or applications.

## 3 One-Round AL via DULO

### 3.1 Algorithm

In this section, we will formalize the problem of one-round AL and present our algorithm to solve the problem via optimizing a learned data utility metric.

**One-round AL problem setup.** Let the feature and the label space be denoted by $\mathcal{X}$ and $\mathcal{Y}$, respectively. A ground-truth labeling function (or the oracle) $f^* : \mathcal{X} \to \mathcal{Y}$ returns the ground-truth label for any given input feature. A one-round AL algorithm is given a pool of unlabeled data points $\mathcal{U} = \{x_i\}$, a set of labeled data points $\mathcal{L} = \{(x_i, f^*(x_i))\}_{i=1}^N$, and a budget $M$. It then selects a subset $S^x$ of size $M$ from the unlabeled pool to request their labels such that the performance of a target model trained on the labeled data is maximized. In particular, the initial labeled data should be small and particularly, insufficient for training a high-performance model.

We define a *data utility function* (DUF) as a mapping from an unlabeled *set* to a real number indicating the utility of the set. We follow a typical AL goal and consider the performance of the model trained on the unlabeled data after labeling as its utility score. Formally, let $S^x = \{x_i\}_{i=1}^n$ denote a set of features. A learning algorithm $\mathcal{A}$ is a (potentially randomized) function that takes a training set $S = \{(x_i, f^*(x_i))\}_{i=1}^n$ as input and returns a classifier $f$. A metric function $u$ takes a classifier as input and outputs its performance. $u(f)$ usually measures $f$'s ability to generalize, such as validation loss or accuracy. Given $\mathcal{A}$, $u$, and $f^*$, a DUF is defined as $U_{\mathcal{A},u,f^*}(S^x) = \mathbb{E}_{\mathcal{A}} \left[ u(\mathcal{A}(\{(x, f^*(x)) | x \in S^x\})) \right]$. When context is clear, we omit subscripts and simply write $U(S^x)$.

One can formulate a one-round AL problem as an optimization problem that seeks an unlabeled set maximizing the DUF: $\text{argmax}_{|S^x|=M, S^x \subseteq \mathcal{U}} U(S^x)$ where $\mathcal{U}$ is the unlabeled pool and $M$ is the labeling budget. Note that each evaluation of $U(S^x)$ requires labeling $S^x$ and training on the labeled instances. Therefore, an unlabeled set selection problem cannot be solved by evaluating the DUF on different choices of unlabeled sets and picking the best one. We propose to train a parametric model $\widehat{U}$ (referred to as a *data utility model* hereinafter) to approximate DUFs based on the initial labeled data. Overall, the proposed algorithm, DULO, proceeds in two stages: (1) learning $\widehat{U}$ from initial labeled data and (2) optimizing $\widehat{U}$ over the unlabeled pool to select the most useful points.

**Stage 1: Data utility learning.** Learning a DUF consists of two steps: utility sampling and utility model training. The initial labeled data set $\mathcal{L} = \{(x_i, f^*(x_i))\}_{i=1}^N$ is randomly partitioned into a training set $\mathcal{L}_{tr}$ and a validation set $\mathcal{L}_{val}$. In the utility sampling step, we randomly sample subsets $S_t$ from $\mathcal{L}_{tr}$, apply the training algorithm $\mathcal{A}$ on $S_t$ and obtain the model $f_t \sim \mathcal{A}(S_t)$. We then calculate the validation accuracy (or loss) of $f_t$ on $\mathcal{L}_{val}$, which gives $u(f_t)$. Let $S_t^x$ denote the set of features in $S_t$. Then, we could score each $S_t$ with its utility $u(f_t)$. We will refer to the scored subsets $\{(S_t^x, u(f_t))\}_{t=1}^T$ as *utility samples*. In the utility model training step, we will learn a model $\widehat{U}$ to approximate the DUF $U$ using the utility samples. Since $U$ is a set function, its output is invariant to input permutation. There has been extensive work on modeling such functions, either by designing neural network architectures that are permutation-invariant in nature (Zaheer et al., 2017) or imposing permutation invariance as a soft constraint via regularization (Alieva et al., 2021). Our data utility learning framework is flexible to incorporate different modeling strategies for set functions and can benefit from the advances of set function learning. In our experiments, we instantiate the utility model $\widehat{U}$ by DeepSets (Zaheer et al., 2017)—a popular model for set function to showcase the AL result enabled by data utility learning.

**Stage 2: Data utility optimization.** With the learned utility model $\widehat{U}$, we will then solve the following optimization problem to select the most useful unlabeled points:

$$\underset{|S^x|=M, S^x \subseteq \mathcal{U}}{\text{argmax}} \widehat{U}(S^x), \tag{1}$$

The challenges for solving the problem above are three-fold. First, the above problem is a combinatorial optimization problem. Finding the global optimal solution is NP-hard in general. Second and more subtly, when the selection budget $M$ is much larger than the size of the initial labeled set $N$ (which is a typical AL scenario), even an optimal solution to the maximization of $\widehat{U}$ in (1) cannot ensure selecting the truly useful data (i.e., the one that maximizes $U$). During the utility learning stage, the maximal size of the input sets that $\widehat{U}$ could be trained on is $N$. Hence, $\widehat{U}$ cannot generalize to unseen unlabeled sets of size much larger than $N$ as effectively as those sets of size less than $N$. Thus, the solution to (1) may not be close to the solution to $\mathrm{argmax}_{|S^x|=M, S^x \subseteq \mathcal{U}} U(S^x)$. Third, evaluation time of $\widehat{U}$ increases with the input size.

To tackle these challenges, we propose a Blockwise Stochastic Greedy (BSG) algorithm. Specifically, at each iteration, we first sample (without replacement) a subset $\mathcal{B} \subset \mathcal{U}$ (called a *block*) from the unlabeled pool $\mathcal{U}$. The block size $B = |\mathcal{B}|$ should be chosen such that the $\widehat{U}$ has both good generalizability and efficient evaluation. We select $\frac{MB}{|\mathcal{U}|}$ data points from each block. Within each block, we run a stochastic greedy algorithm, which also proceeds iteratively. Specifically, in each sub-iteration, we randomly select a subset $Z \subseteq \mathcal{B}$ (without replacement) and find the data point $e$ within $Z$ such that $e$ achieves the highest marginal contribution to unlabeled set $R$ that is already selected *within* $\mathcal{B}$, i.e., $e = \mathrm{argmax}_{e \in Z} \widehat{U}(R \cup \{e\}) - \widehat{U}(R)$. The runtime of BSG is $\mathcal{O}(|\mathcal{U}|)$ in terms of the number of evaluations of $\widehat{U}$. In Section 5.3.5, we conduct an ablation study of how varying block sizes influence active learning performance, and we empirically validate that $B = |\mathcal{L}_{tr}|$ is a great heuristic to ensure both computational efficiency and high data utility for practical use.

The pseudo-code of DULO is provided in Appendix A. It is worth noting that although unlabeled data is selected block-by-block, DULO is completely different from multi-round batch-mode AL, since it does not require the data points selected in the previous blocks to be labeled before we select data points from the current one. Therefore, data selection within different blocks is fully parallelizable.

## 3.2 Computational Considerations

Since our goal is to develop a one-round AL tool for realistic ML tasks, scalability is crucial. While the optimization part of DULO can be made scalable via the BSG algorithm, the computation time of data utility learning still remains a major hurdle. Specifically, constructing a size-$T$ set of utility samples requires retraining a target model for $T$ times. In our ablation study (Section 5.3.3), we found that for DUF to be learned accurately, the number of utility samples required (i.e., the number of target model retraining required) is of the order of the size of the initial labeled data. The overhead of retraining increases with **(1)** the size of the retrained model and **(2)** the size of sets that are trained on.

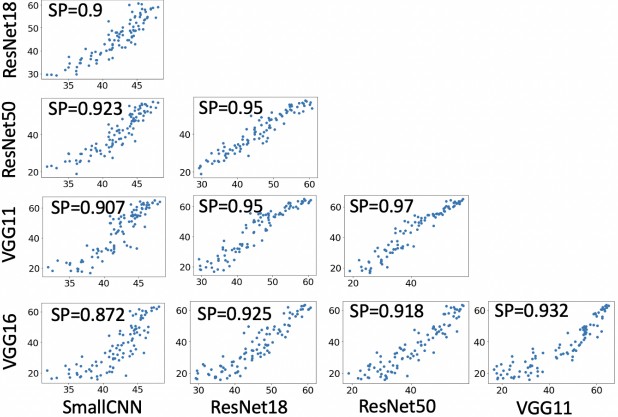

Figure 2: Example of the correlation between data utilities for commonly used models for CIFAR10. The values in both x- and y-axis are test accuracy. 'SP' stands for Spearman's correlation coefficient.

To improve the efficiency of data utility learning for large target models, we propose to use a separate, less computationally intensive *proxy* model in place of the much larger and more accurate target model to generate utility samples. Besides, retraining of a proxy model could be further accelerated by parallelization without any communication overhead. This proxy-model-based idea builds on an important phenomenon observed in many past works (Lewis & Catlett, 1994; Coleman et al., 2019; Nath et al., 2021) that the points that are more useful for one learning algorithm are usually also more useful for another, or in other words, the relative data utility scores of different data subsets are *transferable* between models. Figure 2 illustrates such utility transferability, where the $x$ and $y$-axis of each point represent the accuracy of two different models trained on a given subset and different points are trained on different subsets. Although an expensive classifier's accuracy is much higher than a less expensive classifier, the accuracies of two

classifiers are positively correlated. The observation of utility transferability has been also leveraged to boost efficiency in other applications, including coreset selection (Lewis & Catlett, 1994; Coleman et al., 2019), hyperparameter tuning (Nath et al., 2021) and data valuation (Jia et al., 2019). Despite its wide adoption, how to measure utility transferability remains an open question (Wu & Zhang, 2021). In Section 4, we will rigorously characterize the transferability and present ways to measure it, which could further facilitate the selection of proxy models.

### 3.2.1 Do We Need To Learn the Utility for Large Subsets?

As discussed earlier, the inefficiency of data utility learning also grows with the size of subsets used for retraining a proxy model. However, as these subsets are randomly selected from the initial labeled data and the initial labeled data are often limited in practice, the size of these subsets being trained on is usually small. The readers may worry that if the initial labeled dataset is large, one needs to perform many slow retrainings on many large subsets. In this section, we show that this is not necessary as the variance of utility over different subsets of a given size diminishes as the size increases.

Recent empirical observations (Hestness et al., 2017; Rosenfeld et al., 2019; Kaplan et al., 2020; Hashimoto, 2021) show when the data size is large, the generalization error of a neural network is strikingly predictable with merely the information of data size. The relationship between error and size consistently follows a power-law. Despite the past works on the scaling law phenomenon, they have been mostly focused on an empirical study of how the *average* model performance over different subsets of a given size changes as the size increases. However, in the context of active learning, we are interested in the *confidence interval* of the performance prediction. If the confidence interval is narrow for a given data size, it means that different subsets of that size would result in similar model performance. In that case, it does not matter which subsets of that size are selected and therefore, we do not need to perform active learning at the size.

In this section, we take a step towards analyzing the confidence interval of scaling law-based performance prediction and present the first analysis for a class of deep neural networks whose output behavior closely mimic linear models. Remarkably, this class includes pretrained networks as well as sufficiently wide deep nets (Achille et al., 2021; Mu et al., 2020; Lee et al., 2019). In these scenarios, a deep neural network can be transformed into an equivalent linear model trained with a simple quadratic loss but still reaches a performance similar to the original model. Therefore, we can approximate deep neural networks' DUFs by analyzing the equivalent linear model. Specifically, given a model $f_{\mathbf{w}}(\mathbf{x})$, let $\mathbf{w}_0$ denote an initial set of weights (e.g., pre-trained on ImageNet for image classification task). Following Achille et al. (2021), we consider a linearization $f_{\mathbf{w}}^{\text{lin}}(\mathbf{x})$ of the network $f_{\mathbf{w}}(\mathbf{x})$ given by the first-order Taylor expansion of $f_{\mathbf{w}}(\mathbf{x})$ around $\mathbf{w}_0$: $f_{\mathbf{w}}^{\text{lin}}(\mathbf{x}) = f_{\mathbf{w}_0}(\mathbf{x}) + \nabla_{\mathbf{w}} f_{\mathbf{w}_0}(\mathbf{x}) \cdot (\mathbf{w} - \mathbf{w}_0)$. If the weights $\mathbf{w}$ do not move too much from the initial pre-trained weights $\mathbf{w}_0$ during fine-tuning, then $f_{\mathbf{w}}^{\text{lin}}(\mathbf{x})$ will remain a good approximation of the network while becoming linear in $\mathbf{w}$ (but still remaining highly non-linear with respect to the input $\mathbf{x}$). Effectively, this is equivalent to training a linear classifier using the gradients $\mathbf{z}_i := \nabla_{\mathbf{w}} f_{\mathbf{w}_0}(\mathbf{x}_i)$ as features (Mu et al., 2020). (Achille et al., 2021) showed that equivalent performance can be obtained by replacing the loss function with the regularized least-squares loss $\sum_{i=1}^{n} \left\| f_{\mathbf{w}}^{\text{lin}}(\mathbf{x}_i) - f^*(\mathbf{x}_i) \right\|^2 + \lambda \|\mathbf{w}\|^2$. The advantage of this equivalent formulation of neural networks is that the optimal weights $\mathbf{w}^*$ can now be written in closed-form.

**Theorem 3.1.** *The DUFs for linearized neural network with mean squared error (MSE) metric trained on a i.i.d. sampled dataset $S$ of size $n$ whose gradient distribution $\nabla_{\mathbf{w}} f_{\mathbf{w}_0}(\mathbf{x}_i)$ is subgaussian follows $U(S) = n^{-1}C + O\left(n^{-3/2}\sqrt{\log(1/\delta)}\right)$ with probability at least $1 - \delta$ over the choice of $S$ for sufficiently large $n$. $C$ is a constant depending on test data.*

**Implication of Theorem 3.1 for Data Utility Learning.** The term $O\left(n^{-3/2}\sqrt{\log(1/\delta)}\right)$ in Theorem 3.1 shows that the confidence interval shrinks as the data size grows larger. That is, the performance variation between different subsets become neglectable in large-data regime and hence it is not crucial to learn the difference and choose among different subsets. This implies that we can safely focus data utility learning on the small-data regime, and performing data utility learning on relatively small subsets is a win-win for both efficiency and data selection effectiveness.

# 4 Measuring Utility Transferability

Data utility's transferability across different learning algorithms is an intuitive concept that has been employed to boost computational efficiency in numerous applications that require model retraining (Lewis & Catlett, 1994; Coleman et al., 2019; Jia et al., 2019; Nath et al., 2021). However, to the best of our knowledge, the notion of transferability has not been formally defined before. In this section, we delve into this concept and introduce its first formalization. Subsequently, we present two heuristics aimed at estimating the transferability between two learning algorithms. These guidelines are useful in selecting appropriate proxy models. Specifically, given a set of candidate proxy models, all within a predetermined computational constraint (i.e., the clock time of retraining for multiple times is appropriate), one can pick the model with the highest (estimated) utility transferability against the target model.

**Formalization.** For a given dataset $\mathcal{D} := \{(x_i, f^*(x_i))\}_{i=1}^n$ and a binary vector $\alpha \in \{0,1\}^n$, write $\mathcal{D}_\alpha := \{(x_i, f^*(x_i)) | \alpha_i = 1\}$ as the subset of $\mathcal{D}$ that is selected by the non-zero entries in $\alpha$. Let $w_\alpha := \mathcal{A}(\mathcal{D}_\alpha)$, representing the model obtained by applying the learning algorithm $\mathcal{A}$ on $D_\alpha$. In this section, we assume $\mathcal{A}$ is deterministic for simplicity and readability. Therefore, $U(\{x_i | \alpha_i = 1\}) = u(w_\alpha)$. We start by formally defining utility transferability.

**Definition 4.1.** Given a dataset $\mathcal{D}$, we say two learning algorithms $\mathcal{A}$ and $\widetilde{\mathcal{A}}$ are *p-utility transferable* if $\Pr_{\alpha,\beta}[T(\alpha, \beta) > 0] \geq p$ where

$$T_{\mathcal{A}_1, \mathcal{A}_2}(\alpha, \beta; \mathcal{D}) = (u(w_\beta) - u(w_\alpha)) \cdot (u(\widetilde{w}_\beta) - u(\widetilde{w}_\alpha)) \tag{2}$$

and $(\alpha, \beta) \sim \text{Unif}(\{0,1\})^n \times \text{Unif}(\{0,1\})^n$.

The parameter $p$ reflects the 'extent' of transferability. When $p = 1$, $w$ and $\widetilde{w}$ have perfect utility transferability. We describe two efficient methods to measure transferability. The uniform distribution in the definition results from the Maximum Entropy Principle. Although a proxy model can preserve the target model's selection result as long as the maximal utility subset on the proxy model is also the maximal utility subset on the target model, we do not know the maximal utility subsets *a priori*. Hence, the uniform distribution over pairs of subsets is the most reasonable prior in Definition 4.1. When the context is clear, we omit the learning algorithm pairs $\mathcal{A}_1, \mathcal{A}_2$ and dataset $\mathcal{D}$, and simply write $T(\alpha, \beta)$.

## 4.1 Two Heuristics for Estimating Utility Transferability

A naive of comparing the transferability of different proxy models with a target model is by training the target and proxy models on all possible data subsets and computing $p$ by calculating the fraction of $(\alpha, \beta)$-pairs that satisfy (2). This naive approach is clearly infeasible as both the target and proxy models need to be retrained on exponentially many subsets. Here, we propose two heuristics to evaluate the transferability. These heuristics can be used as a guideline for selecting proxy models.

**Vanilla Monte Carlo (VMC) Method.** One natural idea is to use a Monte Carlo approach to estimate $p$. We can randomly sample $m$ subsets $\mathcal{D}_{\alpha_1}, \ldots, \mathcal{D}_{\alpha_m}$, train the target model and proxy models on each of $\mathcal{D}_{\alpha_i}$, and then count the number of $(i, j)$-pairs such that $T(\alpha_i, \alpha_j) > 0$, $i, j = 1, \ldots, m$. The Monte Carlo estimation of $p$ is thus $\widehat{p} = \frac{\sum_{1 \leq i < j \leq m} I[T(\alpha_i, \alpha_j) > 0]}{m(m-1)/2}$, which is unbiased with sample variance $Var(\widehat{p}) = p(1-p)/(m(m-1)/2))$. Suppose that we are given a pool of $K$ proxy model candidates, then the proxy model selection time is roughly $m \times (\sum_{k=1}^K T_{\text{proxy},k} + T_{\text{target}})$, where $T_{\text{proxy},k}$ and $T_{\text{target}}$ are average computation time for training $k$-th candidate proxy model and the target model, respectively. Proxy models are often chosen such that the training time is neglectable compared to the target model. So the computational time of proxy model selection is dominated by $mT_{\text{target}}$, i.e., retraining the target model for $m$ times. When $p = 1/2$, if we want $Var(\widehat{p}) < 10^{-4}$ (so the standard deviation $< 10^{-2}$), $m \approx 75$. On the other hand, directly learning the data utility model without the proxy model technique requires training the target model for thousands of times. Hence, leveraging a proxy model for data utility learning and (if allowed) spending extra computational budget selecting the most suitable proxy model is in general more recommendable than directly retraining the target model for the sake of efficiency.

**Lower Bound Estimation Based (LBEB) Method.** We develop another Monte-Carlo based approach to select proxy models from the perspective of the lower bound of $p = \Pr_{\alpha,\beta}[T(\alpha,\beta) > 0]$. First notice that $\Pr_{\alpha,\beta}[T(\alpha,\beta) > 0] = \frac{1}{2^n}\sum_{\alpha \in \{0,1\}^n} \Pr_\beta[T(\alpha,\beta) > 0]$. Therefore, we can sample $\mathcal{D}_{\alpha_1}, \dots, \mathcal{D}_{\alpha_m}$ as before and try to bound $\sum_{\alpha_i} \Pr_\beta[T(\alpha_i,\beta) > 0]$. By first-order Taylor expansion, we have $u(w_\beta) - u(w_\alpha) \approx \frac{\partial u}{\partial \alpha}(\beta - \alpha) = \frac{\partial u}{\partial w_\alpha}\frac{\partial w_\alpha}{\partial \alpha}(\beta - \alpha)$. Let $z = \beta - \alpha$. We have $(u(w_\beta) - u(w_\alpha)) \cdot (u(\widetilde{w}_\beta) - u(\widetilde{w}_\alpha)) \approx \left(\frac{\partial u}{\partial w_\alpha}\frac{\partial w_\alpha}{\partial \alpha}z\right) \cdot \left(\frac{\partial u}{\partial \widetilde{w}_\alpha}\frac{\partial \widetilde{w}_\alpha}{\partial \alpha}z\right) = \left(\frac{\partial u}{\partial w_\alpha}\frac{\partial w_\alpha}{\partial \alpha}z\right)^T \cdot \left(\frac{\partial u}{\partial \widetilde{w}_\alpha}\frac{\partial \widetilde{w}_\alpha}{\partial \alpha}z\right) = z^T \left(\frac{\partial w_\alpha}{\partial \alpha}\right)^T \left(\frac{\partial u}{\partial w_\alpha}\right)^T \left(\frac{\partial u}{\partial \widetilde{w}_\alpha}\right) \left(\frac{\partial \widetilde{w}_\alpha}{\partial \alpha}\right) z$. Let $\Sigma = \left(\frac{\partial w_\alpha}{\partial \alpha}\right)^T \left(\frac{\partial u}{\partial w_\alpha}\right)^T \left(\frac{\partial u}{\partial \widetilde{w}_\alpha}\right) \left(\frac{\partial \widetilde{w}_\alpha}{\partial \alpha}\right)$.

If the learning algorithm $w_\alpha$ is ERM style gradient-based algorithm, then $\left(\frac{\partial w_\alpha}{\partial \alpha}\right)^T$ can be computed by implicit function theorem (similar to the technique used in (Koh & Liang, 2017)). $\Pr_z[z^T\Sigma z > 0]$ indicates the degree of transferrability between two models. Since we are particularly interested in the transferability between two models of different sizes, $\Sigma$ is asymmetric. But note that $z^T\Sigma z > 0$ if and only if $z^T(\Sigma + \Sigma^T)z > 0$. Hence, we can instead analyze the symmetric matrix $A := \Sigma + \Sigma^T$. We derive a lower bound of the probability for which $z^T A z > 0$.

**Theorem 4.2.** *For any fixed $\alpha \in \{0,1\}^n$, let $S = (\beta - \alpha)^T A(\beta - \alpha)$. We have $\Pr_{\beta \sim \text{Unif}(\{0,1\})^n}[S > 0] \geq 1 - \exp\left(\frac{-\mu^2}{2\sum_{k=1}^n c_k^2}\right)$ where $c_k$ and $\mu$ are a function of $A$ and $\alpha$ and can be computed in $O(k)$ and $O(n)$ runtime, respectively.*

The exact computation of $c_k$ and $\mu$ is deferred to Appendix B.2 due to space constraints. From this theorem, we know that the lower bound of $\Pr_{\beta \sim \text{Unif}(\{0,1\})^n}[z^T A z > 0]$ can be thus characterized by the ratio $C(A(\alpha), \alpha) = \frac{\mu^2}{\sum_{k=1}^n c_k^2}$, and the quantity $\sum_{i=1}^m C(A(\alpha_i), \alpha_i)$ can be used as an efficient measure for transferability. A larger value of this quantity implies stronger transferability.

We compare the effectiveness of these two approaches in Section 5.2.

## 5 Evaluation

In this section, we first evaluate the two building blocks of DULO: (1) the effectiveness of using data utility model to predict the performance of a model trained on a given dataset, and (2) the effectiveness of the two heuristics proposed in Section 4 for measuring the transferability between learning algorithms. Finally, we evaluate the performance of DULO and compare it with the existing one-round AL strategies and the state-of-the-art batch AL strategies in one-round setting.

### 5.1 Performance of Data Utility Prediction

We empirically evaluate the quality of data utility prediction from a trained DeepSet-based utility model. Specifically, we randomly sample data subsets of the same size from the initial labeled training data $\mathcal{L}_{tr}$ ('seen data') and unlabeled pool $\mathcal{U}$ ('unseen data'), and test the performance of the trained data utility model $\widehat{U}$ in predicting the test accuracy of proxy models trained on the subset. As Figure 3 shows, the predicted test accuracy is highly correlated with actual test accuracy. The mean square error of the prediction for unseen data is $< 0.0024$ for MNIST dataset and $< 10^{-4}$ for CIFAR10 dataset.

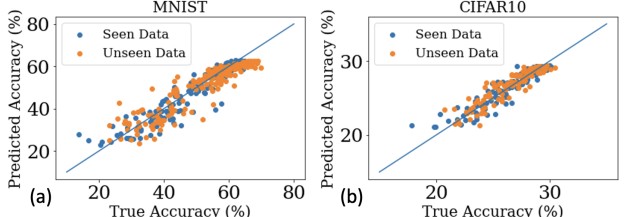

Figure 3: Prediction of the trained data utility model on validation subsets. "Seen data" indicates the elements in the validation subsets are used during utility model training; "unseen" indicates otherwise.

We also note that $\widehat{U}$ will slightly underestimate the utility of high-quality dataset and overestimate the utility of low-quality dataset. However, the overall trend is preserved, as the Spearman coefficient is 0.933 for MNIST and 0.913 for CIFAR10. Implementation details and additional results on more datasets are shown in Appendix D.2.

## 5.2 Estimating Utility Transferability

In Figure 4, we empirically evaluate the effectiveness of VMC and LBEB proposed in Section 4 to measure transferability. Specifically, for 8 most commonly used models for CIFAR10 and MNIST, we train them on the same $m$ subsets of the full datasets, compute $\widehat{p} = \frac{\sum_{1 \le i < j \le m} I[T(\alpha_i, \alpha_j) > 0]}{m(m-1)/2}$ and $\sum_{i=1}^{m} C(A(\alpha_i), \alpha_i)$ for each *pair* of models. We set $m \le 90$ in the experiment. To establish the ground truth, we train 8 different models on 5000 ($\gg 90$) subsets of MNIST/CIFAR10, and rank each pair of models according to $\widehat{p}$ computed over $m = 5000$. We then rank each pair of models according to the VMC and the LBEB method, and the performance of transferability measurements is evaluated by the number of misordered pair of model pairs, with respect to the ground truth transferability ranking. As we can see, both methods are able to achieve a low misorder rate ($< 18\%$) when $m \ge 60$. For small $m$, LBEB method achieves better performance in measuring transferability.

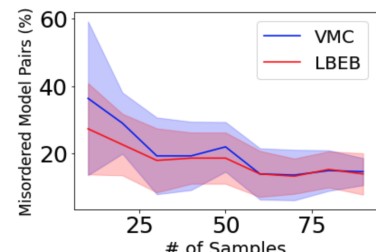

Figure 4: VMC method vs. LBEB method to compare transferability between models. The error bar indicates the standard deviation across 10 runs of random set sampling.

## 5.3 Active Learning

**Protocol.** We evaluate the performance of different AL strategies on different types of models and a varied amount of selected data points. We evaluate the robustness of DULO on different practical scenarios such as imbalanced or noisy unlabeled datasets. We also test the performance on real-world datasets and provide insights into the selected data. Finally, we perform ablation studies for the sample complexity and hyperparameters for DULO. A potential variant for DULO that is combined with Uncertainty Sampling is deferred to Appendix D.5.1.

**Datasets & Proxy Models & Implementation.** We summarize the dataset and proxy/target model settings in Table 1. The proxy models are selected based on the tradeoff between the utility transferability with the target model and computational budget. All of the $\mathcal{L}_{tr}$ and $\mathcal{U}$ are sampled from the corresponding datasets' original training data, where the sampling distribution is uniform unless otherwise specified. For all datasets, we randomly sample 4000 subsets of $\mathcal{L}_{tr}$ and use the corresponding proxy model to generate the training data for utility learning. We set stochastic greedy optimization's precision parameter $\varepsilon = 10^{-3}$ and optimization block size $B = 2000$ in all experiments. For each experiment setting, we repeat DULO and other baseline algorithms 10 times to obtain the average AL performance and error bars. We defer the dataset descriptions, model architectures, hyperparameters and other experiment details to Appendix D.

**Choosing proxy models.** The proxy models in our experiments are shown in Table 1 were judiciously chosen by weighing both utility transferability with the target model and computational feasibility. For each dataset, we assessed two proxy model architectures that are sufficiently different for a more meaningful comparison. From these, our selection favored the model that offers a superior balance between transferability and computational efficiency. To provide greater clarity, we also include the alternative proxy models considered. When both proxy models displayed similar computational runtime or both were comfortably within our computational constraints, we naturally gravitated towards the model exhibiting superior transferability. In cases where one model might have shown marginally better transferability but came at a considerably steeper computational price (for instance, ResNet18 for CIFAR10), we opted for the model that ensured a more practical runtime.

**Baselines.** We consider the state-of-the-art batch AL as well as one-round AL strategies as our baselines: **(1) FASS** (Wei et al., 2015) first filters out the data samples with low uncertainty about predictions. It then selects a subset by first assigning each unlabeled data a hypothesized label with the partially-trained classifier's prediction and optimizing a Nearest Neighbor submodular function on the unlabeled dataset with hypothesized labels. **(2) BADGE** (Ash et al., 2019) first generates hypothesized labels and selects a subset based on the diverse gradient embedding obtained with the hypothesized samples. **(3) GLISTER** (Killamsetty et al., 2020) also generates hypothesized labels and performs discrete bi-level optimization problem on the hypothesized samples. **(4) Coreset** (Sener & Savarese, 2017) tackles active learning through the lens of core-set selection.

| Dataset | $|\mathcal{L}_{tr}|$ | $|\mathcal{L}_{val}|$ | Selected Proxy Model (time / $\widehat{p}$) | Alternative Proxy Model (time / $\widehat{p}$) | Target Model (time) |
|---|---|---|---|---|---|
| **MNIST** | 300 | 300 | LR (0.75 hrs / 0.832) | KNN (0.1 hrs / 0.718) | LeNet (1.5 hrs) |
| **CIFAR10** | 500 | 500 | SmallCNN (0.9 hrs / 0.824) | MLP (0.4 hrs / 0.778) | Small VGG (12 hrs) |
| **USPS** | 300 | 300 | SVM-RBF (1 min / 0.794) | SVM-Linear (1 min / 0.752) | LR (8 min) |
| **PUBFIG83** | 500 | 500 | LeNet (1 hrs / 0.872) | MLP (0.6 hrs / 0.697) | Small VGG (18 hrs) |
| **CIFAR100** | 1000 | 1000 | SmallCNN (2 hrs / 0.844) | ResNet18 (11 hrs / 0.848) | ResNet50 (1.5 day) |
| **IMDb** | 1000 | 1000 | Small RNN (1 day / 0.8) | Bag of Word + KNN (0.4 hrs / 0.63) | LSTM (12 days) |

Table 1: Dataset and Model Settings. The 'time' in the table means the rough clock time for training 4000 proxy/target models. $\widehat{p}$ means the proxy model's estimated utility transferability with the target model using LBEB method from Section 4.

| | Unbalanced Dataset | | | | Noisy Dataset | | | |
|---|---|---|---|---|---|---|---|---|
| | MNIST | | CIFAR10 | | MNIST | | CIFAR10 | |
| **Labeling Budget** | **500** | **1000** | **5000** | **10000** | **500** | **1000** | **5000** | **10000** |
| **DULO** | 89.89 (0.39) | 92.36 (0.30) | 52.97 (1.03) | 60.81 (0.35) | 90.36 (0.27) | 91.03 (0.10) | 58.49 (1.00) | 66.11 (0.36) |
| **Random** | 88.83 (0.53) | 90.79 (0.56) | 49.82 (0.16) | 57.86 (1.23) | 88.12 (0.43) | 89.56 (0.23) | 41.93 (5.97) | 46.86 (7.82) |
| **FASS** | 88.82 (0.43) | 91.11 (0.32) | 48.17 (3.06) | 57.85 (2.50) | 88.24 (0.61) | 90.41 (0.83) | 52.12 (0.56) | 61.45 (1.14) |
| **BADGE** | 88.99 (0.37) | 91.00 (0.33) | 48.82 (2.09) | 56.51 (1.32) | 87.23 (0.54) | 89.03 (0.34) | 55.35 (1.16) | 64.38 (0.18) |
| **GLISTER** | 88.89 (0.62) | 91.37 (0.50) | 47.56 (1.68) | 55.14 (0.54) | 86.54 (0.46) | 89.68 (0.75) | 54.05 (0.93) | 60.90 (2.97) |
| **CoreSet** | 89.20 (0.31) | 90.39 (0.15) | 51.14 (0.81) | 55.78 (0.57) | 89.28 (0.11) | 89.68 (0.12) | 53.07 (1.34) | 62.53 (0.74) |
| **TED** | 88.51 (0.15) | 89.54 (0.13) | - | - | 87.79 (0.27) | 89.11 (0.15) | - | - |
| **EBM** | 88.42 (0.09) | 89.64 (0.03) | - | - | 88.08 (0.16) | 88.46 (0.34) | - | - |

Table 2: Performance comparison of DULO with other baselines on imbalanced and noisy datasets. For each labeling budget, we plot the accuracy (in %) of models trained on the selected dataset. Standard errors are shown in '()'.

Contrary to relying on hypothesized labels, this method leverages an upper bound, independent of data labels, to substitute the conventional coreset loss. **(5) Transductive Experiment Design (TED)** (Yu et al., 2006) selects the samples that minimize the expected predictive variance of ridge regression on the data. **(6) Error Bound Minimization (EBM)** (Gu et al., 2012) derive a deterministic, label-independent out-of-sample error bound for Laplacian regularized Least Squares trained on subsampled data, and select a subset of data points to label by minimizing the derived upper bound. **(7) Random** is a setting where we randomly select a subset from the unlabeled pool.

FASS, BADGE, GLISTER, and Coreset are initially designed for multi-round AL, but they can be extended to the one-round setting by only performing 1 round of data selection. The implementation details for the baselines are deferred to Appendix D.4.

*Remark* 5.1. Since the considered baselines such as BADGE and GLISTER have been shown to outperform simple techniques like uncertainty sampling (Settles, 2009), we do not compare them in this work. Moreover, we do not compare with the one-round AL technique proposed in Jin et al. (2022) as it is specialized to the specific application. We do not compare with the one-round AL technique proposed in Shoham & Avron (2020) as its implementation has not been released, and it is uncertain how the Gram matrix of the infinite NTK is computed based on the paper's description.

### 5.3.1 Results on Imbalanced and Noisy Dataset

We artificially generate class-imbalance for MNIST unlabeled dataset by sampling 55% of the instances from one class and the rest of the instances uniformly across the remaining 9 classes. For CIFAR10's unlabeled dataset, we sample 50% of the instances from two classes, 25% instances from another two classes, and 25% instances from the remaining 6 classes. The results for MNIST and CIFAR10 dataset are shown in Table 2. As we can see, DULO significantly outperforms the other baselines. By examining the data points selected by different strategies, we found that DULO can indeed select more balanced dataset, thereby leading to higher performance.

To create noisy datasets, we inject Gaussian noise into images in random subsets of data. The noise scale refers to the standard deviation of the random Gaussian noise. For MNIST, we add each of the noise scales

|  | **DULO** | **Random** | **FASS** | **BADGE** | **GLISTER** | **CoreSet** |
|---|---|---|---|---|---|---|
| **USPS** | 88.43 (0.28) | 87.56 (0.41) / 0.005 | 87.59 (0.41) / 0.0 | 87.00 (0.60) / 0.0 | 87.38 (0.52) / 0.0 | 87.663 (0.396) / 0.002 |
| **PubFig83** | 51.55 (0.66) | 50.87 (1.01) / 0.027 | 49.29 (2.95) / 0.100 | 50.29 (1.16) / 0.003 | 51.5 (2.18) / 0.014 | 50.837 (0.586) / 0.003 |
| **CIFAR100** | 77.18 (0.42) | 76.02 (0.46) / 0.003 | 75.91 (0.99) / 0.005 | 76.2 (1.27) / 0.037 | 73.44 (1.38) / 0.000 | 76.165 (0.518) / 0.0 |
| **IMDb** | 88.25 (0.44) | 87.95 (0.52) / 0.098 | 88.12 (0.2) / 0.179 | 88.13 (0.45) / 0.259 | 88.13 (0.15) / 0.216 | 87.843 (0.308) / 0.015 |

Table 3: Datasets with Natural Quality Variations. The selection budget for USPS, PubFig83, CIFAR100 and IMDb are 1000, 2000, 25000 and 20000, respectively. The p-values for paired t-test for the comparison between DULO and each of the baseline are shown after '/'. Notably, the majority of these p-values are below the 0.05 threshold, signifying that DULO's improvement over the baselines is statistically significant in most scenarios.

of 0.25, 0.6, and 1.0 to 25% unlabeled data (i.e., 75% of the unlabeled data are noisy and there are 3 different noisy levels). For CIFAR10, we add noise to 25% unlabeled data points with noise scale 1.0. Table 2 shows that DULO outperforms other baselines by more than 1% for every data selection size, and is very close to, or sometimes even better the performance on a randomly selected clean dataset. To get more insights into the selection process, we inject an MNIST image together with its three noisy variants of different noise scales into the MNIST unlabeled dataset and examine their selection order in DULO, shown in Figure 5 (a). DULO tends to select clean, high-quality images early and is able to sift out noisy data.

### 5.3.2 Results on Datasets with Natural Quality Variations

We study the effectiveness of different AL strategies on datasets from different domains, including USPS, PubFig83, CIFAR100, and IMDb dataset. These datasets usually contain some natural variations in data quality. The results are shown in Table 3. We can see that DULO consistently outperforms all other baselines in almost all of the settings and all labeling budgets. Figure 5 (b) illustrates the images selected in different ranks by DULO. We can see that the points selected earlier have higher image quality; especially, the last five selected '6' digit images tend to be blurry and some could easily be confused with other digits. The last ten selected images contain four '1's, which is because USPS is a class-imbalanced dataset, and there are more '1's than other classes.

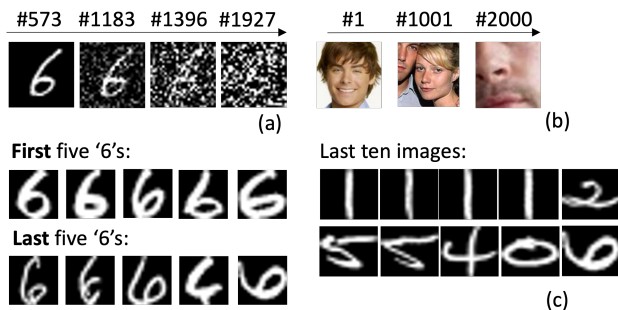

Figure 5: Selection ranks for different images in a greedy optimization block. (a) MNIST, (b) PubFig83, (c) USPS.

Figure 5 (c) shows the rankings of 3 images within an optimization block. As we can see, the early face images selected by DULO contain complete face features, while the later ones either contain more irrelevant features or are corrupted. Additional experiments on MNIST and CIFAR10 can be found in Appendix D.5.4.

### 5.3.3 Ablation Study of Size of Utility Samples

The overhead of data utility learning hinges on the size of utility samples for training proxy models. Figure 6 shows the one-round AL performance for data utility models trained with different amounts of utility samples. We find that the performance is nearly optimal as long as the number of utility samples is above 500, which only takes around 12 min to generate. Therefore, a practical guideline for setting the size of initial labeled dataset is 10-50 data points per class, and for setting the number of utility samples is at least 2× the total number of initial labeled dataset.

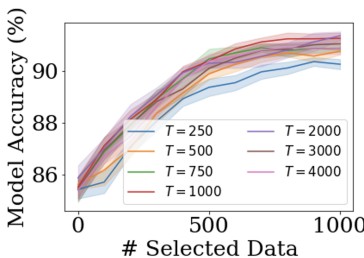

Figure 6: Effect of # utility samples on DULO performance.

### 5.3.4 Robustness to Scarcity of initial labeled Data

Table 4 shows the one-round AL performance for utility models trained with different amount of initial labeled training data points $|\mathcal{L}_{tr}|$. Note that here the largest possible training samples for the proxy model varies, but for fair comparison, we still train the classifiers with 300 initial labeled training samples together with selected data points. We fix the block size and number of subsets sampled for utility training in the experiments. As we can see, near-optimal performance could be achieved even when there are only 100 labeled training data points. When $|\mathcal{L}_{tr}|$ is too large, the trained utility model's performance might degrade due to insufficient training samples compared with input size. When $|\mathcal{L}_{tr}|$ is too small, the

| Labeling Budget | 500 | 1000 |
|---|---|---|
| **DULO** with $|\mathcal{L}_{tr}| = 50$ | 86.15 (0.39) | 87.22 (0.31) |
| **DULO** with $|\mathcal{L}_{tr}| = 75$ | 88.07 (0.28) | 88.79 (0.30) |
| **DULO** with $|\mathcal{L}_{tr}| = 100$ | 90.36 (0.21) | 91.16 (0.27) |
| **DULO** with $|\mathcal{L}_{tr}| = 200$ | 90.26 (0.27) | 91.26 (0.21) |
| **DULO** with $|\mathcal{L}_{tr}| = 300$ | 90.36 (0.27) | 91.03 (0.10) |
| **DULO** with $|\mathcal{L}_{tr}| = 500$ | 90.63 (0.31) | 91.12 (0.32) |
| **DULO** with $|\mathcal{L}_{tr}| = 1000$ | 89.88 (0.26) | 91.00 (0.20) |

Table 4: Performance of DULO with utility models trained with varying sizes of initial labeled training data $|\mathcal{L}_{tr}|$.

generalizability of utility model to large sets would be worse and at the same time, the block size has a small upper limit, which can capture less data interaction.

### 5.3.5 Ablation Study of Optimization Block Size

Figure 7 (a) shows the one-round AL performance with different optimization block sizes $B$. As we can see, both too small and too large block sizes can degrade the performance of one-round active learning. This is because when $B$ is too small, the data utility model fails to capture interactions between data points selected from different blocks. When $B$ is too large, the data utility model has a large generalization error as it never sees such a large dataset during its training time. Although the stochastic optimization algorithm runs in $O(|\mathcal{U}|)$ regardless of the block size (without parallelization), individual DeepSets evaluation time increases significantly with

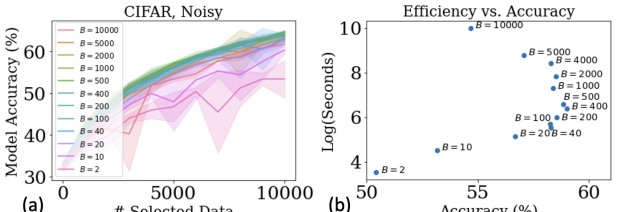

Figure 7: Ablation Study: (a) compares DULO's performance with different optimization block sizes, and (b) studies the relationship between efficiency and AL performance for different block sizes when the number of selected data is 6000.

a larger input set. Figure 7 (b) shows the runtime vs accuracy plot with different block sizes. We can see that when $B$ is too large, it suffers from both poor utility and inefficiency. However, we find a wide range of $B$ that achieves both good efficiency and high data utility (the points located on the right of the figure), which is around $|\mathcal{L}_{tr}|$ ($|\mathcal{L}_{tr}| = 500$ for CIFAR10 in our setting). This servers as a heuristic choice of $B$. In this range, the block size is small enough so that DeepSets models generalize well on input sets of block sizes, while also being large enough so that the utility model could capture most of the data interactions.

## 6 Conclusion & Future Works

This work proposes a general framework for one-round AL, an important AL setting where only one round of label query is allowed, via DUF learning and optimization as well as the use of proxy models. We develop a principled approach for selecting the proxy models. Our evaluation shows that it outperforms existing baselines across various datasets and models.

There are two assumptions underlying our approach: (1) the labels of the initial labeled dataset are correct and (2) the labeled and unlabeled datasets should share the same label space. While these two assumptions usually hold in practice, extending DULO to noisy initial labeled data and varied label settings would be interesting future directions.

## Acknowledgments

This work is supported by Amazon-Virginia Tech Initiative for Efficient and Robust Machine Learning, the National Science Foundation under Grant No. IIS-2312794, NSF IIS-2313130, NSF OAC-2239622, and the Cisco Award. We are grateful to anonymous reviewers at TMLR for their valuable feedback.

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

# A  Pseudocode of the Proposed One-Round AL Algorithm

---

**Algorithm 1:** DULO for One-Round AL

---

**input** : $\mathcal{L}$ - initial labeled data, $\mathcal{A}$ - training algorithm for classifier $f$, $u$ - metric function, $\mathcal{A}_U$ - training algorithm for utility model $\widehat{U}$, $T$ - number of utility samples, $\mathcal{U}$ - unlabeled data pool, $M$ - labeling budget, $B$ - optimization block size, $\varepsilon$ - precision parameter for stochastic greedy algorithm.

**output:** $S_{selected}$ - selected set.

1 Split $\mathcal{L}$ into training set $\mathcal{L}_{tr}$ and validation set $\mathcal{L}_{val}$.
2 // Data Utility Learning
3 Initialize set of utility sample $\mathcal{S}_U = \emptyset$.
4 **for** $t = 1, \dots, T$ **do**
5      Randomly choose a subset $S_t = (S_t^x, f^*(S_t^x)) \subseteq \mathcal{L}_{tr}$.
6      Train classifier $f_t \leftarrow \mathcal{A}(S_t)$ (or $f_t \sim \mathcal{A}(S_t)$ if $\mathcal{A}$ is stochastic).
7      $\mathcal{S}_U = \mathcal{S}_U \cup \{(S_t^x, u(f_t, \mathcal{L}_{val}))\}$.
8 **end**
9 Train $\widehat{U} \leftarrow \mathcal{A}_U(\mathcal{S}_U)$.
10 // Data Utility Optimization
11 Initialize $S_{selected} = \emptyset$, $\mathcal{U}' = \mathcal{U}$.
12 **while** $|S_{selected}| < M$ **do**
13      Sample $\mathcal{B} \subseteq \mathcal{U}'$ of size $B$.
14      $R \leftarrow \emptyset$
15      **while** $|R| < \frac{MB}{|\mathcal{U}|}$ **do**
16          Sample $Z \subseteq \mathcal{B} \setminus R$ of size $\frac{|\mathcal{U}|}{M} \log(1/\varepsilon)$
17          Find $e = \operatorname{argmax}_{e \in Z} \widehat{U}(R \cup \{e\})$
18          $R \leftarrow R \cup \{e\}$
19      **end**
20      $S_{selected} \leftarrow S_{selected} \cup R$.
21      $\mathcal{U}' \leftarrow \mathcal{U}' \setminus R$.
22 **end**
23 **return** $S_{selected}$

---

*Remark* A.1 (Proxy Model Selection). If a pool of candidate proxy models that meet a given computational budget (i.e., efficiently trainable for thousands of times) is available, then we can pick the one with the highest (estimated) utility transferability against the target model. The utility transferability can be estimated by the two methods proposed in Section 4.

*Remark* A.2 (Comparison with Zero-round Active Learning (Chen et al., 2021)). The zero-round AL approach in Chen et al. (2021) is based on leveraging labeled data from related domains (i.e., source domains) which may not necessarily be from the target domain. This requires the existence of such domain-specific labeled datasets. In many practical scenarios, finding a source domain that is sufficiently related to the target domain can be challenging, and domain adaptation might not yield the desired results due to the vast differences between source and target domains. From this perspective, one-round AL is more flexible as it allocates some budget to label a few samples from the target domain directly, thereby ensuring the model has direct information related to the target and isn't wholly reliant on domain adaptation.

## B    Proofs

### B.1    Theorem 3.1

*Theorem* 3.1 (Restated). The DUFs for linearized neural network with mean squared error (MSE) metric trained on a i.i.d. sampled dataset $S$ of size $n$ whose gradient distribution $\nabla_{\mathbf{w}} f_{\mathbf{w}_0}(\mathbf{x}_i)$ is subgaussian follows

$$U(S) = n^{-1}C + O\left(\frac{\sqrt{\log(1/\delta)}}{n^{3/2}}\right) \tag{3}$$

with probability at least $1 - \delta$ over the choice of $S$ for sufficiently large $n$. $C$ is a constant depending on test data.

*Proof.* The proof follows from (Hashimoto, 2021) with the major differences in the regularization term of linear regression. We first change the notation to the traditional one used in linear regression: write $\beta = \mathbf{w} - \mathbf{w}_0 \in \mathbb{R}^d$, $X := \mathbf{Z} \in \mathbb{R}^{n \times d}$, $y := \mathbf{Y} - f_{\mathbf{w}_0}(\mathbf{X}) \in \mathbb{R}^d$, so we have

$$\hat{\beta} = \left(X^T X + \lambda I\right)^{-1} X^T y \tag{4}$$

Assuming $y = X\beta + \varepsilon$ for some $\beta \in \mathbb{R}^d, \varepsilon \sim \mathcal{N}(0, \sigma^2 I)$. Then the covariance of $\hat{\beta} - \beta$ with respect to $\varepsilon$ becomes

$$\mathbb{E}_{\varepsilon}\left[(\beta - \hat{\beta})(\beta - \hat{\beta})^T\right] \tag{5}$$

$$= \mathbb{E}_{\varepsilon}\left[\left(\left(X^T X + \lambda I\right)^{-1} X^T y - \beta\right)\left(\left(X^T X + \lambda I\right)^{-1} X^T y - \beta\right)^T\right] \tag{6}$$

$$= \mathbb{E}_{\varepsilon}\left[\left(X^T X + \lambda I\right)^{-1}\left(X^T y - \left(X^T X + \lambda I\right)\beta\right)\left(X^T y - \left(X^T X + \lambda I\right)\beta\right)^T\left(X^T X + \lambda I\right)^{-1}\right] \tag{7}$$

$$= \mathbb{E}_{\varepsilon}\left[\left(X^T X + \lambda I\right)^{-1}\left(X^T(X\beta + \varepsilon) - \left(X^T X + \lambda I\right)\beta\right)\left(X^T(X\beta + \varepsilon) - \left(X^T X + \lambda I\right)\beta\right)^T\left(X^T X + \lambda I\right)^{-1}\right] \tag{8}$$

$$= \mathbb{E}_{\varepsilon}\left[\left(X^T X + \lambda I\right)^{-1}\left(X^T \varepsilon - \lambda\beta\right)\left(X^T \varepsilon - \lambda\beta\right)^T\left(X^T X + \lambda I\right)^{-1}\right] \tag{9}$$

$$= \left(X^T X + \lambda I\right)^{-1} X^T \mathbb{E}_{\varepsilon}[\varepsilon\varepsilon^T] X\left(X^T X + \lambda I\right)^{-1} \tag{10}$$

$$= \sigma^2 \left(X^T X + \lambda I\right)^{-1} X^T X\left(X^T X + \lambda I\right)^{-1} \tag{11}$$

$$= \sigma^2 \left(X^T X + \lambda I\right)^{-1} - \lambda\sigma^2\left(\left(X^T X + \lambda I\right)^{-1}\right)^2 \tag{12}$$

Denote $A = X^T X + \lambda I$. Assume $\sigma^2 = 1$ for simplicity. If each data point $x_i$ in $X$ are i.i.d. sampled from some subgaussian distribution $x_i \sim p$. Let $X^* \in \mathbb{R}^{m \times d}$ be some test data. Now we want to bound the expected loss

$$L(n) = \frac{1}{m}\mathbb{E}_{X,\varepsilon}\left[\left\|X^*(\beta - \hat{\beta})\right\|_2^2\right] \tag{13}$$

First, by fixing $X$, define $e = \beta - \hat{\beta}$ we can see that

$$\mathbb{E}_{\varepsilon} \left[ \left\| X^*(\beta - \hat{\beta}) \right\|_2^2 \right] \tag{14}$$

$$= \mathbb{E}_{\varepsilon} \left[ tr \left( e^T X^{*T} X^* e \right) \right] \tag{15}$$

$$= \mathbb{E}_{\varepsilon} \left[ tr \left( X^{*T} X^* e e^T \right) \right] \tag{16}$$

$$= tr \left( X^{*T} X^* \mathbb{E}_{\varepsilon}[e e^T] \right) \tag{17}$$

$$= tr \left( X^{*T} X^* \left( A^{-1} - \lambda (A^{-1})^2 \right) \right) \tag{18}$$

$$= tr \left( X^{*T} X^* A^{-1} \right) - \lambda tr \left( X^{*T} X^* (A^{-1})^2 \right) \tag{19}$$

The challenge now is thus to estimate $\mathbb{E}_X[A^{-1}]$ and $\mathbb{E}_X[(A^{-1})^2]$.

Define $\Sigma = \mathbb{E}[X^T X]/n$. Since $X$ is drawn from some subgaussian distribution, by matrix concentration inequality, with probability with at least $1 - \delta$ we have

$$\left\| X^T X n^{-1} - \Sigma \right\|_{op} \leq C \frac{\sqrt{\log(1/\delta)}}{\sqrt{n}} \tag{20}$$

for sufficiently large $n$. Let $\Delta = X^T X n^{-1} - \Sigma$. Therefore $A = n(\Sigma + \Delta) + \lambda I = n\left( (\Sigma + \frac{\lambda}{n}I) + \Delta \right)$
By equality $(P + Q)^{-1} = P^{-1} + \sum_{t=1}^{\infty} P^{-1}(-QP^{-1})^t$, we have

$$A^{-1} = n^{-1} \left( (\Sigma + \frac{\lambda}{n}I) + \Delta \right)^{-1} \tag{21}$$

$$= n^{-1} \left[ (\Sigma + \frac{\lambda}{n}I)^{-1} + \sum_{t=1}^{\infty} (\Sigma + \frac{\lambda}{n}I)^{-1} \left( -\Delta(\Sigma + \frac{\lambda}{n}I)^{-1} \right)^t \right] \tag{22}$$

whenever $\left\| \Delta(\Sigma + \frac{\lambda}{n}I)^{-1} \right\|_{op} < 1$. Since $\left\| \Delta(\Sigma + \frac{\lambda}{n}I)^{-1} \right\|_{op} \leq \left\| \Delta \right\|_{op} \left\| (\Sigma + \frac{\lambda}{n}I)^{-1} \right\|_{op}$, and since $\sigma_{min}(\Sigma + \frac{\lambda}{n}I) \geq \sigma_{min}(\Sigma)$ which is a constant, we know that this series converge for sufficiently large $n$. In this case, the first term is the dominant one.

$$\mathbb{E} \left[ tr \left( X^{*T} X^* A^{-1} \right) \right] \tag{23}$$

$$= tr \left( X^{*T} X^* n^{-1} (\Sigma + \frac{\lambda}{n}I)^{-1} \right) + n^{-1} \mathbb{E} \left[ tr \left( X^{*T} X^* \sum_{t=1}^{\infty} (\Sigma + \frac{\lambda}{n}I)^{-1} \left( -\Delta(\Sigma + \frac{\lambda}{n}I)^{-1} \right)^t \right) \right] \tag{24}$$

where the second term is dominated by

$$tr \left( X^{*T} X^* \sum_{t=1}^{\infty} (\Sigma + \frac{\lambda}{n}I)^{-1} \left( -\Delta(\Sigma + \frac{\lambda}{n}I)^{-1} \right)^t \right) \tag{25}$$

$$\approx tr \left( X^{*T} X^* (\Sigma + \frac{\lambda}{n}I)^{-1} (-\Delta)(\Sigma + \frac{\lambda}{n}I)^{-1} \right) \tag{26}$$

$$\leq d \left\| X^{*T} X^* \right\|_{op} \left\| (\Sigma + \frac{\lambda}{n}I)^{-1} \right\|_{op}^2 \left\| \Delta \right\|_{op} \tag{27}$$

$$= O \left( \frac{\sqrt{\log(1/\delta)}}{\sqrt{n}} \right) \tag{28}$$

So

$$\mathbb{E}\left[tr\left(X^{*T}X^{*}A^{-1}\right)\right] = tr\left(X^{*T}X^{*}n^{-1}(\Sigma + \frac{\lambda}{n}I)^{-1}\right) + O\left(\frac{\sqrt{\log(1/\delta)}}{n^{3/2}}\right) \tag{29}$$

$$= n^{-1}tr\left(X^{*T}X^{*}(\Sigma + \frac{\lambda}{n}I)^{-1}\right) + O\left(\frac{\sqrt{\log(1/\delta)}}{n^{3/2}}\right) \tag{30}$$

By a similar argument, we have

$$\mathbb{E}\left[tr\left(X^{*T}X^{*}(A^{-1})^{2}\right)\right] = n^{-2}tr\left(X^{*T}X^{*}\left((\Sigma + \frac{\lambda}{n}I)^{-1}\right)^{2}\right) + O\left(\frac{\sqrt{\log(1/\delta)}}{n^{3}}\right) \tag{31}$$

Therefore

$$L(n) \tag{32}$$

$$= \frac{1}{m}\left[n^{-1}tr\left(X^{*T}X^{*}(\Sigma + \frac{\lambda}{n}I)^{-1}\right) - n^{-2}tr\left(X^{*T}X^{*}\left((\Sigma + \frac{\lambda}{n}I)^{-1}\right)^{2}\right) + O\left(\frac{\sqrt{\log(1/\delta)}}{n^{3/2}}\right)\right] \tag{33}$$

Set $\lambda = n\lambda'$, we have

$$L(n) = \frac{1}{m}\left[n^{-1}tr\left(X^{*T}X^{*}(\Sigma + \lambda'I)^{-1}\right) + O\left(\frac{\sqrt{\log(1/\delta)}}{n^{3/2}}\right)\right] \tag{34}$$

under the condition of

$$\left\|X^{T}Xn^{-1} - \Sigma\right\|_{op} \le C\frac{\sqrt{\log(1/\delta)}}{\sqrt{n}} \tag{35}$$

where this event happens with probability at least $1 - \delta$. So the expected loss has diminishing return as $n$ grows with high probability for sufficiently large $n$. □

## B.2  Theorem 4.2

*Theorem* 4.2 (Restated). For any fixed $\alpha \in \{0,1\}^{n}$, let $S = (\beta - \alpha)^{T}A(\beta - \alpha)$. We have

$$\Pr_{\beta \sim \text{Unif}(\{0,1\})^{n}}[S > 0] \ge 1 - \exp\left(\frac{-\mu^{2}}{2\sum_{k=1}^{n}c_{k}^{2}}\right) \tag{36}$$

where $c_{k}$ and $\mu$ are a function of $A$ and $\alpha$ and can be computed in $O(k)$ and $O(n)$ runtime, respectively.

*Proof.* Since $\beta \sim \text{Unif}(\{0,1\})^{n}$, therefore $\beta_{i} - \alpha_{i} \sim \text{Unif}(\{0,1\})$ if $\alpha_{i} = 0$, and $\beta_{i} - \alpha_{i} \sim \text{Unif}(\{-1,0\})$ if $\alpha_{i} = 1$. Let $x = \beta - \alpha$. Denote $S_{k} = \sum_{i,j=1}^{k} x_{i}A_{ij}x_{j}$, so

$$S_{n} = x^{T}Ax = \sum_{i=1}^{n}\sum_{j=1}^{n} x_{i}A_{ij}x_{j} \tag{37}$$

We first note that $\mu = \mathbb{E}[S_n]$ can be easily computed as

$$\mu = \mathbb{E}[S_n] \tag{38}$$

$$= \mathbb{E}\left[\sum_{i=1}^{n}\sum_{j=1}^{n} x_i A_{ij} x_j\right] \tag{39}$$

$$= \sum_{i=1}^{n} A_{ii}\,\mathbb{E}\left[x_i^2\right] + \sum_{i \neq j} A_{ij}\,\mathbb{E}\left[x_i\right]\mathbb{E}\left[x_j\right] \tag{40}$$

$$= \frac{1}{4}\sum_{i=1}^{n} A_{ii} + \sum_{i \neq j} A_{ij}\,\mathbb{E}\left[x_i\right]\mathbb{E}\left[x_j\right] \tag{41}$$

Define

$$Y_k = S_k - \mathbb{E}\left[S_k\right] \tag{42}$$

Then it's easy to see that for all $k \geq 0$ we have

$$\mathbb{E}\left[Y_{k+1}|Y_k\right] = Y_k \tag{43}$$

so the sequence of $(Y_0, Y_1, \ldots, Y_n)$ is a martingale. Also,

$$|Y_{k+1} - Y_k| \tag{44}$$

$$= |S_{k+1} - S_k - \mathbb{E}\left[S_{k+1} - S_k\right]| \tag{45}$$

$$= \left|a_{k+1,k+1}x_{k+1}^2 + 2x_{k+1}\sum_{i=1}^{k} a_{i,k+1}x_i - \mathbb{E}\left[a_{k+1,k+1}x_{k+1}^2 + 2x_{k+1}\sum_{i=1}^{k} a_{i,k+1}x_i\right]\right| \tag{46}$$

$$= \left|a_{k+1,k+1}(x_{k+1}^2 - 1/2) + 2x_{k+1}\sum_{i=1}^{k} a_{i,k+1}x_i - \mathbb{E}\left[2x_{k+1}\sum_{i=1}^{k} a_{i,k+1}x_i\right]\right| \tag{47}$$

$$= \left|x_{k+1}\left(a_{k+1,k+1}x_{k+1} + 2\sum_{i=1}^{k} a_{i,k+1}x_i\right) - \left(a_{k+1,k+1}/2 + 2\,\mathbb{E}[x_{k+1}]\sum_{i=1}^{k} a_{i,k+1}\,\mathbb{E}[x_i]\right)\right| \tag{48}$$

Let

$$c_k = \sup_{x_1,\ldots,x_k}\left|x_k\left(a_{k,k}x_k + 2\sum_{i=1}^{k-1} a_{i,k}x_i\right) - \left(\frac{1}{2}a_{k,k} + 2\,\mathbb{E}[x_k]\sum_{i=1}^{k-1} a_{i,k}\,\mathbb{E}[x_i]\right)\right| \tag{49}$$

Since everything is linear, $c_k$ could be efficiently computed in $O(k)$ as

$$b_k = \mathbb{E}\left[\frac{1}{2}a_{k,k} + 2\,\mathbb{E}[x_k]\sum_{i=1}^{k-1} a_{i,k}\,\mathbb{E}[x_i]\right] \tag{50}$$

$$c_k = \max\left(|b_k|,\right. \tag{51}$$

$$\left|F(x_k)\left(a_{k,k}F(x_k) + 2\sum_{i=1}^{k-1}\max(0, a_{i,k}F(x_i))\right) - b_k\right|, \tag{52}$$

$$\left.\left|F(x_k)\left(a_{k,k}F(x_k) + 2\sum_{i=1}^{k-1}\min(0, a_{i,k}F(x_i))\right) - b_k\right|\right) \tag{53}$$

where $F(x_k) = 1$ if $x_k \sim \text{Unif}(\{0,1\})$, and $F(x_k) = -1$ if $x_k \sim \text{Unif}(\{-1,0\})$.

By Azuma's inequality, we have

$$\Pr\left[Y_n \leq -\varepsilon\right] = \Pr\left[S_n - \mathbb{E}[S_n] \leq -\varepsilon\right] \leq \exp\left(\frac{-\varepsilon^2}{2\sum_{k=1}^{n} c_k^2}\right) \tag{54}$$

By setting $\varepsilon = \mathbb{E}[S_n]$, we have

$$\Pr\left[x^T A x \leq 0\right] \leq \exp\left(\frac{-\mathbb{E}[S_n]^2}{2\sum_{k=1}^{n} c_k^2}\right) \tag{55}$$

$\square$

We note that the value of $c_k$ depends on the order of rows/columns of $A$. Therefore, in practice, we can further improve this bound by randomly sample many different permutations and get the smallest value.

## C    Stochastic Greedy Algorithm

For completeness, we briefly introduce the stochastic greedy algorithm (SG) from (Mirzasoleiman et al., 2015) here, and refer the readers to the original work for a more thorough introduction. The stochastic greedy algorithm is a simple approach that, for each iteration, randomly selects a subset of data and then finds the best data point within that subset. This approach was originally proposed for maximizing submodular functions. In the context of optimizing utility model (i.e., the trained DeepSets model), the "best data point" within each randomly selected subset refers to the data point $z$ with the highest marginal contribution $\widehat{U}(S \cup \{z\}) - \widehat{U}(S)$, where $S$ is the set of data points selected in previous iterations. The pseudo-code is outlined in Algorithm 2.

The main distinction between this approach and the vanilla greedy algorithm is that the candidate data points to be selected in each iteration of stochastic greedy algorithm is a smaller, randomly selected subset instead of all unselected data points. Thus, this approach is more efficient than vanilla greedy optimization and the runtime is linear in the number of dataset size to achieve $1 - 1/e - \varepsilon$ optimization guarantee for monotone submodular functions ($\varepsilon$ is a precision parameter).

Although SG provides no theoretical approximation guarantee for general functions, our experiments show that it achieves high empirical performance on DUFs. We conjecture that this is because DUFs usually exhibits diminishing return property similar to submodular functions. Exploring different approaches for optimizing data utility models are interesting future directions.

---

**Algorithm 2:** Stochastic Greedy Optimization for Utility Model

---

**input** : dataset $\mathcal{D}$, trained utility model $\widehat{U} : 2^{\mathcal{D}} \to \mathbb{R}$, target selection size $k$, precision parameter for stochastic greedy algorithm $\varepsilon$.

**output:** A set $S \subseteq \mathcal{D}$ s.t. $|S| = k$.

**1** $S \leftarrow \emptyset$.

**2 for** $t = 1, \ldots, k$ **do**

**3**    Sample $R \subseteq \mathcal{D} \setminus S$ of size $\frac{|\mathcal{D}|}{k} \log(1/\varepsilon)$

**4**    Find $z = \operatorname{argmax}_{z \in R} \widehat{U}(S \cup \{z\})$

**5**    $S \leftarrow S \cup \{z\}$

**6 end**

**7 return** $S$

---

# D  Additional Experiment Settings

## D.1  Transferability Evaluation

Figure 2 shows examples of the transferability of data utilities between different learning algorithms. The dataset here we use is CIFAR10. For each point in the figures, we randomly sample 5000 CIFAR10 images and add Gaussian noise to a random portion of the images. We then record the test accuracies of different model architectures trained on the dataset.

For the experiment in Section 4 that compares VMC and LBEB in measuring transferability, we use 5 models on CIFAR10 and 3 models on MNIST. Specifically, for CIFAR10 we use ResNet18, ResNet50, VGG11, VGG16, and a small CNN model is used as the proxy model for CIFAR10 dataset, which has two convolutional layers, each is followed by a max pooling layer and a ReLU activation function. For MNIST we use logistic regression, LeNet and a slightly larger CNN model which has six convolutional layers. There are $\binom{5}{2} + \binom{3}{2} = 13$ model pairs in total, and we want to rank the transferability of these 13 model pairs. we train them on the same $m$ subsets of the full datasets, and compute $\widehat{p} = \frac{\sum_{1 \le i < j \le m} I[T(\alpha_i, \alpha_j) > 0]}{m(m-1)/2}$ and $\sum_{i=1}^{m} C(A(\alpha_i), \alpha_i)$ for each *pair* of models. We set $m \le 90$ in the experiment. To establish the ground truth, we train 8 different models on 5000 ($\gg 90$) subsets of MNIST/CIFAR10, and rank each pair of models according to $\widehat{p}$ computed over $m = 5000$. We then rank each pair of models according to the VMC and the LBEB method, and the performance of transferability measurements is evaluated by the number of misordered pair of model pairs, with respect to the ground truth transferability ranking.

In the computation of $\sum_{i=1}^{m} C(A(\alpha_i), \alpha_i)$, we need to first compute $\frac{\partial u}{\partial \alpha} = \frac{\partial u}{\partial w_\alpha} \frac{\partial w_\alpha}{\partial \alpha}$. We use the similar trick in influence function technique (Koh & Liang, 2017) based on implicit function theorem, which involves the computation of Hessian inverse $\left( \frac{\partial^2 u}{\partial w_\alpha \partial w_\alpha^T} \right)^{-1}$. Since this may be computationally expensive for large models, we follow the widely used heuristic and only choose to use the last layer of the model. We compute the exact Hessian matrix in our experiment, and the Hessian inverse computation can be further accelerated with techniques such as conjugate gradient (Koh & Liang, 2017). For the computation of $c_k$ and $\mu$, since they depend on the permutation of matrix $A$ and $\alpha$, we randomly sample 2000 permutations and pick the maximum value of $C(A(\alpha_i), \alpha_i)$.

## D.2  Implementation Details for Figure 3 and Additional Results

In Figure 3, all sampled subsets are of the same size, where for MNIST the size is 250 and for CIFAR10 the size is 350. For each subset sampling, we first uniformly randomly generate a real value $\alpha \in [1, 20]$ for each class, and then draw a distribution sample $p$ from Dirichlet distribution $Dir(\alpha_1, \ldots, \alpha_K)$ where $K$ represents the number of classes. We draw a subset with different class sizes proportional to $p$. This sampling design is to ensure the diversity of class distributions in the sampled subsets.

Additional results on CIFAR100 and IMDb dataset are shown in Figure 8. As we can see, these more intricate datasets align with previously observed tendencies: $\widehat{U}$ tends to slightly underestimate the utility for high-quality subsets and overestimate for low-quality subsets. Notwithstanding, the overarching trend remains consistent, with the Spearman correlation coefficient between the predicted and actual model performance is 0.828 for CIFAR100 and 0.86 for IMDb. We conjecture that due to the inherent challenges of these tasks, neural networks might tend to fit a constant value approximating the mean of the target values. Delving deeper into the intricacies of this phenomenon offers an intriguing avenue for future research.

## D.3  Details of Datasets Used in Section 5

**MNIST.**  MNIST consists of 70,000 handwritten digits. The images are $28 \times 28$ grayscale pixels.

**CIFAR10.**  CIFAR10 consists of 60,000 3-channel images in 10 classes (airplane, automobile, bird, cat, deer, dog, frog, horse, ship and truck). Each image is of size $32 \times 32$.

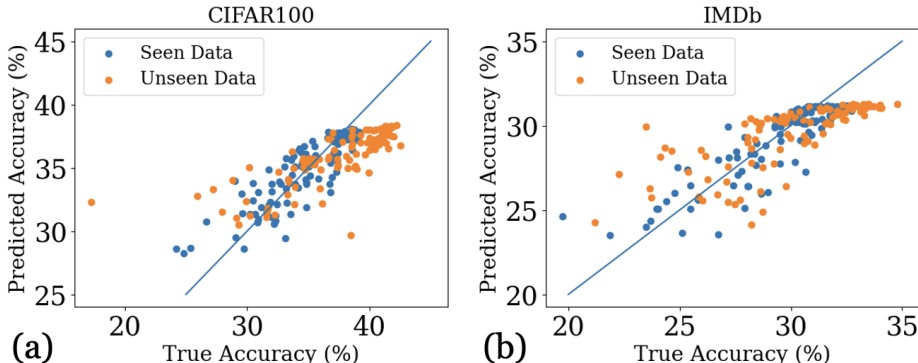

Figure 8: Prediction of the trained data utility model on validation subsets. "Seen data" indicates the elements in the validation subsets are used during utility model training; "unseen" indicates otherwise. The Spearman correlation coefficient between the predicted and actual model performance is 0.828 for CIFAR100 and 0.86 for IMDb.

**USPS.** USPS is a real-life dataset of 9,298 handwritten digits. The images are $16 \times 16$ grayscale pixels. The dataset is class-imbalanced with more 0 and 1 than the other digits.

**PubFig83.** PubFig83 is a real-life dataset of 13,837 facial images for 83 individuals. Each image is resized to $32 \times 32$.

**CIFAR100.** CIFAR100 is similar to CIFAR10 except it has 100 classes.

**IMDB.** The IMDb dataset consists of 50,000 movie reviews, each marked as being a positive or negative review.

### D.4 Implementation Details

**Target Models & Proxy Models.** A small CNN model is used as the proxy model for CIFAR10 and CIFAR100 datasets, which has two convolutional layers, each is followed by a max pooling layer and a ReLU activation function. LeNet model is the proxy model for PubFig83 dataset, and we also used it to test the AL performance for MNIST in the experiment. LeNet is adapted from (LeCun et al., 1998), which has two convolutional layers, two max pooling layers and one fully-connected layer. A mini VGG model[2] is used to evaluate AL performance on CIFAR10 and PubFig83, which has six convolutional layers, and each of them is followed by a batch normalization layer and a ReLU layer. We use Adam optimizer with learning rate $10^{-3}$, mini-batch size 32 to train all of the aforementioned models for 30 epochs, except that we train LeNet for 5 epochs when using it for testing AL performance on MNIST. We also use the support vector machine (SVM) to evaluate AL performance on the USPS dataset. We implement SVM with scikit-learn (Pedregosa et al., 2011) and set the L2 regularization parameter to be 0.1. For IMDb, we use LSTM model follows from PyTorch tutorial[3] as the target model, which use a pretrained word embedding. A smaller RNN model adapted from the same source is used as the proxy model. All other training details follow from the same tutorial.

All of our experiments are performed on Tesla K80 GPU.

**DeepSets as Architecture for Utility Learning.** A DeepSets model is a function $\widehat{U}(S) = \rho(\sum_{x \in S} \phi(x))$ where both $\rho$ and $\phi$ are neural networks. In our experiment, both $\phi$ and $\rho$ networks have 3 fully-connected layers. For MNIST and USPS, we set the number of neurons to be 128 in each hidden layer. For CIFAR10

---

[2]https://github.com/microsoft/LQ-Nets/blob/master/cifar10-vgg-small.py
[3]https://github.com/bentrevett/pytorch-sentiment-analysis

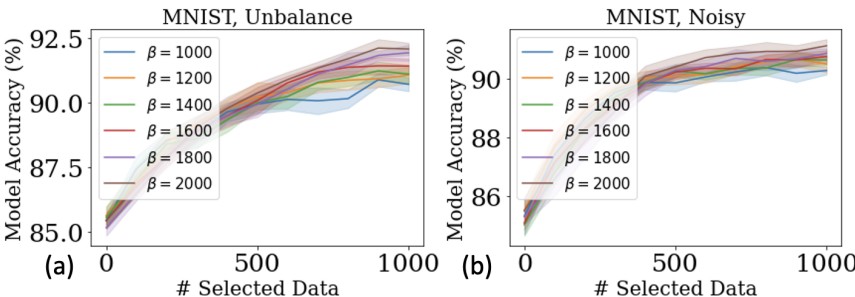

Figure 9: Ablation Study: the performance of DULO when we run greedy optimization on the $\beta$ most uncertain samples of the current classifiers.

and PubFig83, we set the number of neurons in every hidden layer to be 512. We set the dimension of set features (i.e., the output of $\phi$ network) for DeepSets models to be 128 in all experiments, except for USPS dataset the set feature number is 64. We use the Adam optimizer with learning rate $10^{-5}$, mini-batch size of 32, $\beta_1 = 0.9$, and $\beta_2 = 0.999$ to train all of the DeepSets-based utility models. For IMDb dataset, the utility learning as well as baseline algorithms is performed on the embedding of the corresponding sentence encodings.

The size of the utility samples is chosen uniformly at random from the training set. This design is because in stochastic greedy optimization, the utility model needs to predict the utility for subsets of all sizes smaller than the block size. It is interesting future work to investigate whether a different sampling strategy can further improve performance.

**Baseline Implementation.** In terms of the baseline batch AL approaches, we set $\beta$ in FASS to be the size of unlabeled dataset after parameter tuning. We set the learning rate in GLISTER to be 0.05, following the original paper. We test these baselines with open-source implementation[4]. The model performance is evaluated on the hold-out test set associated with each dataset. For TED algorithm, we follow the original paper and set the coefficient for $\ell_2$ regularizer as 0.1. For EBM algorithm, we set the coefficient for $\ell_2$ regularizer and manifold regularization as 0.1 and 0.01, respectively.

### D.5 Additional Results

#### D.5.1 Combining $DULO$ with Uncertainty Sampling

Traditionally, AL techniques like uncertainty sampling (US) and query by committee (QBC) have shown great promise in several domains of machine learning (Settles, 2009). However, in the task of multi-round batch AL, naive uncertainty sampling fails to capture the interactions between selected samples. Simply choosing the most uncertain samples may lead to a selected set with very low diversity. Filtered Active Submodular Selection (FASS) (Wei et al., 2015) combines the uncertainty sampling method with a submodular data subset selection framework. Specifically, at every round $t$, FASS first selects a candidate set of $\beta_t$ most uncertain samples among unlabeled data, and then runs greedy optimization on an appropriate submodular objective (with the hypothesized labels assigned by the current model). It is natural to ask whether we should also combine DULO with uncertainty sampling by only performing greedy optimization on the most uncertain samples. We evaluate the performance of DULO when we first filter out data points that the current classifier has high confidence about, and preserve a candidate set of $\beta$ most uncertain samples on which we run stochastic greedy optimization. We show the performance of different values of $\beta$ on class-imbalanced and noisy MNIST dataset in Figure 9, where $\beta = |\mathcal{U}| = 2000$ coincides with the setting of the original DULO algorithm. As we can see, for all settings studied in our experiments, $\beta = 2000$ consistently outperforms other smaller values of $\beta$. Hence, using uncertainty sampling to pre-process unlabeled samples does not seem to lead to better performance in our one-round AL framework. We conjecture that this is because we only

---

[4]https://github.com/decile-team/distil

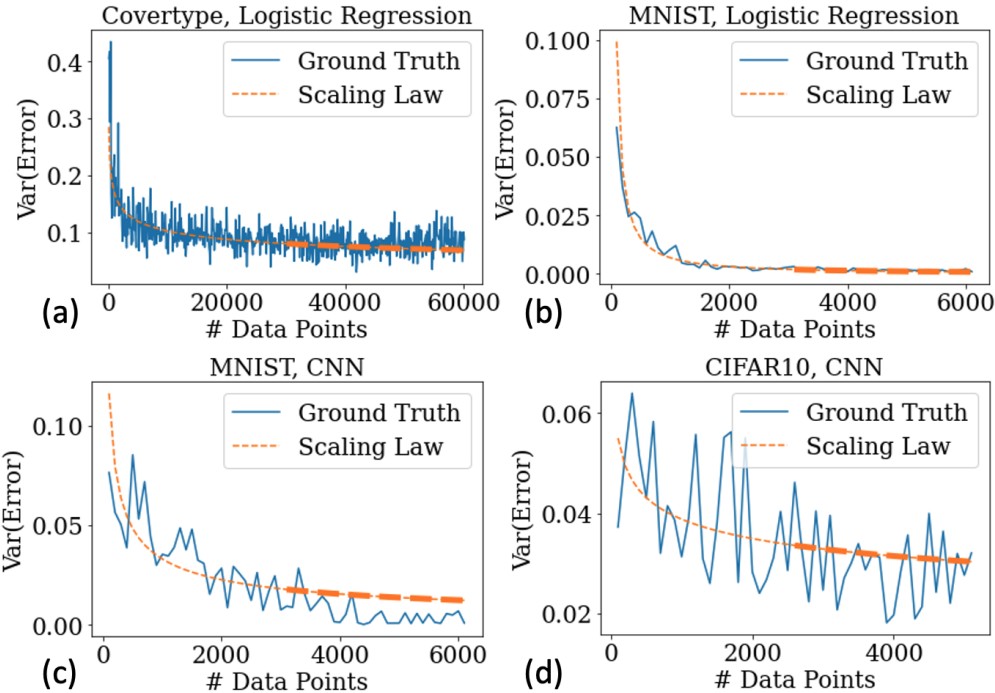

Figure 10: Illustration of the scaling law prediction for different datasets and different architectures. The heavier orange dashed line corresponds to the variance of model test loss *predicted* by the scaling law. The "error" in the y-axis refers to cross entropy loss. In the figure, "CNN" refers to a small CNN model adapted from PyTorch tutorial `https://pytorch.org/tutorials/beginner/blitz/cifar10_tutorial.html`.

have very few labeled samples at the beginning. In that case, the performance of the initial classifiers is pretty poor and their uncertainty outputs are not informative.

### D.5.2 Do We Need To Learn DUF in Large Data Regime? Predict Utility Variance with Scaling Law

In this section, we show the practical implication of our Section 3.2.1 and Theorem 3.1. Many recent literature on the Scaling Law of Model Performance (Hestness et al., 2017; Rosenfeld et al., 2019; Kaplan et al., 2020; Hashimoto, 2021) propose to predict model performance on the large data regime by fitting a function $\log(\text{error}) \approx -\alpha \log(n) + C$, where $n$ is the dataset size, and $\alpha$ and $C$ are parameters to be fitted. Our Theorem 3.1 suggests that there may also be certain scaling law style relationship between the dataset size and the variance of model performance, where the randomness is due to the i.i.d. sampling of dataset from a natural distribution. ' Thus, we can predict the variance of model performance on the large data regime by first fitting a function $\log(\text{Var}(\text{error})) \approx -\alpha \log(n) + C$ in the small data regime (where training a model is fast), and then extrapolating the prediction to large data regime (where training a model is slow). Based on the prediction, one can determine whether it is worth performing active learning for a given data acquisition budget. If the predicted utility variability is small, then active learning strategies may only bring small improvements upon random sampling. In that case, it could be not that worthwhile performing active learning.

**Experiment.** We wish to scale datasets while preserving the original distribution. Thus, given a dataset $D$, we subsample subsets whose cardinalities are $n = r|D|$, where $r = 0.1\%, 0.2\%, \ldots, 9.9\%, 10\%$. For each $r$, we subsample 10 subsets uniformly at random from the original dataset. We train a model on each of the subsets and compute the variance of model loss on a hold-out test set for a given cardinality. We then fit a function $\log(\text{Var}(\text{error})) \approx -\alpha \log(n) + C$ on the cardinalities that are of $0.1\%, 0.2\%, \ldots, 4.9\%, 5\%$ of the original dataset, and use the learned function to predict the utility variance for cardinalities that are of $5.1\%, 5.2\%, \ldots, 9.9\%, 10\%$ of the original dataset. Figure 10 shows the result of scaling law prediction on three different

| | Unbalanced Dataset | | | | Noisy Dataset | | | |
|---|---|---|---|---|---|---|---|---|
| | MNIST | | CIFAR10 | | MNIST | | CIFAR10 | |
| Labeling Budget | 500 | 1000 | 5000 | 10000 | 500 | 1000 | 5000 | 10000 |
| DULO - Selected Proxy Model | 89.89 (0.39) | 92.36 (0.30) | 52.97 (1.03) | 60.81 (0.35) | 90.36 (0.27) | 91.03 (0.10) | 58.49 (1.00) | 66.11 (0.36) |
| DULO - Alternative Proxy Model | 88.83 (0.16) | 91.57 (0.25) | 50.62 (0.58) | 59.34 (0.34) | 89.20 (0.42) | 90.58 (0.04) | 56.92 (0.35) | 65.51 (0.43) |
| Random | 88.83 (0.53) | 90.79 (0.56) | 49.82 (0.16) | 57.86 (1.23) | 88.12 (0.43) | 89.56 (0.23) | 41.93 (5.97) | 46.86 (7.82) |
| FASS | 88.82 (0.43) | 91.11 (0.32) | 48.17 (3.06) | 57.85 (2.50) | 88.24 (0.61) | 90.41 (0.83) | 52.12 (0.56) | 61.45 (1.14) |
| BADGE | 88.99 (0.37) | 91.00 (0.33) | 48.82 (2.09) | 56.51 (1.32) | 87.23 (0.54) | 89.03 (0.34) | 55.35 (1.16) | 64.38 (0.18) |
| GLISTER | 88.89 (0.62) | 91.37 (0.50) | 47.56 (1.68) | 55.14 (0.54) | 86.54 (0.46) | 89.68 (0.75) | 54.05 (0.93) | 60.90 (2.97) |
| CoreSet | 89.20 (0.31) | 90.39 (0.15) | 51.14 (0.81) | 55.78 (0.57) | 89.28 (0.11) | 89.68 (0.12) | 53.07 (1.34) | 62.53 (0.74) |
| TED | 88.51 (0.15) | 89.54 (0.13) | - | - | 87.79 (0.27) | 89.11 (0.15) | - | - |
| EBM | 88.42 (0.09) | 89.64 (0.03) | - | - | 88.08 (0.16) | 88.46 (0.34) | - | - |

Table 5: Performance comparison of DULO with other baselines on imbalanced and noisy datasets where we include the performance of DULO when using alternative proxy models. For each labeling budget, we plot the accuracy (in %) of models trained on the selected dataset. Standard errors are shown in '()'.

datasets (Covertype (Blackard, 1998), MNIST, CIFAR10) with different model architectures. As we can see, the scaling law generally performs well for predicting the variance of model test loss in the large data regime.

**Discussion.** Overall, the consequence of Section 3.2.1 and Theorem 3.1 is two-fold. First, it provides a theoretical understanding of data utility function (DUF): the data utility becomes more predictable by the size of training data when the size gets larger. Thus, in AL, the choice of unlabeled data sets matters more when the selection budget is smaller. Second and more importantly, it provides practical guidance for designing the input size of data utility model in one-round AL. The computational overhead of data utility learning grows with the size of subsets used for model retraining. The result in Section 5 shows that performing data utility learning on relatively small subsets is a win-win for both efficiency and data selection effectiveness (as the utility of larger sets has much less variance).

### D.5.3 Evaluate Alternative Proxy Models

To emphasize the advantage of selecting proxy models with higher utility transferability, we additionally evaluate the performance of DULO when using the alternative proxy model shown in Table 1, and we show the result in Table 5. The results clearly show that DULO, when paired with the selected proxy model, outperforms its counterpart with the alternative model. Notably, even when using the alternative proxy model, DULO's performance remains competitive with other baselines.

### D.5.4 Experiments on Regular setting of MNIST and CIFAR10

The one-round AL performance on MNIST and CIFAR10 without any data corruption is shown in Table 6. Intriguingly, in this setting, the performance across all methods, including the random selection baseline, is quite close to each other. We conjecture that this is because of the high curation quality of datasets like MNIST and CIFAR10. Nevertheless, DULO consistently demonstrated a competitive performance compared with other baselines for most of the cases.

### D.5.5 Runtime

We report the running clock time for all baselines we use in Table 7. While DULO is not as efficient as multi-round AL techniques, it aligns with our expectations for a one-round AL approach. Given the intrinsic nature of one-round AL, it demands more computations and greater information extraction from the initial labeled dataset. However, it's worth highlighting that DULO holds a marked computational edge over traditional one-round AL methods. This is because these earlier techniques necessitate the computation of the inverse of the Fisher information matrix, a process that is highly computationally heavy for high-dimensional data.

| | MNIST | | CIFAR10 | |
|---|---|---|---|---|
| **Labeling Budget** | **500** | **1000** | **5000** | **10000** |
| **DULO** | 96.47 (0.11) | 96.89 (0.08) | 61.72 (1.71) | 68.98 (0.62) |
| **Random** | 95.23 (0.17) | 96.43 (0.22) | 61.31 (1.71) | 68.25 (0.61) |
| **FASS** | 94.73 (0.05) | 96.45 (0.22) | 61.70 (1.74) | 69.03 (0.64) |
| **BADGE** | 94.34 (0.29) | 96.86 (0.08) | 60.78 (1.67) | 68.77 (0.56) |
| **GLISTER** | 95.99 (0.18) | 96.49 (0.07) | 61.10 (1.66) | 69.24 (0.64) |
| **CoreSet** | 96.45 (0.11) | 96.94 (0.08) | 61.03 (1.69) | 69.51 (0.58) |
| **TED** | 95.13 (0.20) | 96.07 (0.20) | - | - |
| **EBM** | 95.22 (0.12) | 96.39 (0.24) | - | - |

Table 6: Performance comparison of DULO with other baselines on clean datasets. For each labeling budget, we plot the validation accuracy (in percentage) of models trained on the selected dataset. Standard errors are shown in '()'.

| | MNIST | | CIFAR10 | |
|---|---|---|---|---|
| **Labeling Budget** | 500 | 1000 | 5000 | 10000 |
| **DULO** | 0.75 hrs | 0.75 hrs | 0.9 hrs | 0.9 hrs |
| **FASS** | 6.474 | 6.621 | 33.147 | 69.283 |
| **BADGE** | 19.39 | 37.163 | 336.722 | 636.26 |
| **GLISTER** | 5.236 | 12.707 | 35.336 | 67.158 |
| **CoreSet** | 2.652 | 3.566 | 38.641 | 62.645 |
| **TED** | 289.436 | 1170.006 | 1 day + | 1 day + |
| **EBM** | 286.618 | 1201.159 | 1 day + | 1 day + |

Table 7: Average Runtime (in seconds) for DULO and different baselines in selecting a fixed amount of data points on MNIST and CIFAR10 dataset, with the setting described in Section 5.3.1. The clock time is recorded when running on an NVIDIA A100 80GB GPU and an AMD 64-Core CPU Processor.

