# OpenReview forum: "One-Round Active Learning through Data Utility Learning and Proxy Models"
_TMLR — Accepted by TMLR_

### Review · Reviewer_hRdt · 2023-08-06

**Summary Of Contributions:**

This work presents a novel general framework for one-round active learning (AL), an essential AL setting where only a single round of label query is allowed. The framework is based on data utility function (DUF) learning and optimization techniques, along with the utilization of proxy models. Notably, a principled approach for selecting the proxy models is developed.

**Audience:**

Yes

**Broader Impact Concerns:**

This submission does not contain Broader Impact Statement.

**Claims And Evidence:**

Yes

**Requested Changes:**

Required change are list here:

- In the experiments, the authors still conduct experiments very similar to multi-round AL settings like Figure 6, line chart is not appropriate for this task since it looks like multi-round AL settings, should use tables instead;
- Add more baselines that close to one-round AL setting;
- Discuss the difference between this submission and [r1];
- Add more experiments on standard data settings;
- The proposed method is not a great improvement compared with the baselines in Table 2, it's better to add a paired t-test to evaluated if the proposed model is significantly better than baselines.


**Strengths And Weaknesses:**

Strength: The idea of one-round AL setting is pretty interesting and useful, and the findings of Figure 2 about the utility transferability would aslo be meaningful, the accuries of shallow DNNs and deeper DNNs are positive correlated, which could help reduce the computational cost in AL sampling processes.

Weakness: The experimental settings of this paper are somewhat not reasonable.
- The baselines' chosen: The authors have already discussed the shortcomings of multi-round AL like BADGE, FASS, GLISTER in Related Work part, said these methods suffer unsatisfactory performance in the early rounds as the classifier used for generating hypothesized labels is trained on limited labeled data. The authors also mentioned some AL methods related one-round AL. However, in experiments, the authors only adopted multi-round AL strategies like BADGE, FASS, GLISTER as baselines and force them to use one-round setting, which is unfair. The authors should compare their proposed methods with baselines that closer to one-round settings.
- This papers has overlap with zero-round AL [r1], the author should discuss the differences between this two papers and illustrate in the case that there is already a zero-round AL model, the existence of one-round AL is necessary.
- Although noisy and imbalance data settings are more challenge, it is still need to compare the AL methods on standard data settings.


[r1] Chen S, Wang T, Jia R. Zero-Round Active Learning[J]. arXiv preprint arXiv:2107.06703, 2021.

---

> ### Author Response · Authors · 2023-09-02
>
> **Q1 [Change line charts to tables & additional baselines of earlier one-round AL techniques]**
>
> **A:** We sincerely appreciate the constructive feedback from the reviewer. We have also incorporated earlier one-round AL methods for kernel regression models, including Transductive Experiment Design (TED) [1] and Error Bound Minimization (EBM) [2] as additional baselines. We would like to highlight that these methods exhibited considerable computational overhead. Specifically, they did not reach completion within a 24-hour timeframe on CIFAR10 dataset; hence, we present their results solely on the MNIST dataset (that was also the largest dataset these approaches worked in their original papers). This is because these earlier techniques necessitate the computation of the inverse of the Fisher information matrix, a process that is highly computationally heavy for high-dimensional data. We do not compare with the one-round AL technique proposed in [3] as it is specialized to the specific task of image segmentation. We do not compare with the one-round AL technique proposed in [4] as its implementation has not been released, and it is uncertain how the Gram matrix of the infinite NTK is computed based on the paper's description.
>
> In light of the recommendation of using tables instead of line charts, we have converted most of our results representation from line charts to tables. As can be observed from Table 2 and subsequent results, DULO consistently maintains an edge over these additional baselines, underlining its effectiveness.
>
> [1] Yu, Kai, Jinbo Bi, and Volker Tresp. "Active learning via transductive experimental design." Proceedings of the 23rd international conference on Machine learning. 2006.
>
> [2] Gu, Quanquan, et al. "Selective labeling via error bound minimization." Advances in neural information processing systems (2012).
>
> [3] Jin, Qiuye, et al. "One-shot active learning for image segmentation via contrastive learning and diversity-based sampling." Knowledge-Based Systems 241 (2022): 108278.
>
> [4] Shoham, Neta, and Haim Avron. "Experimental design for overparameterized learning with application to single shot deep active learning." IEEE Transactions on Pattern Analysis and Machine Intelligence (2023).
>
>
> **Q2 [Difference compared with [r1]]**
>
> **A:** We appreciate the reviewer drawing attention to the work on zero-round AL [r1]. The zero-round AL approach in [r1] is based on leveraging labeled data from related domains (i.e., source domains) which may not necessarily be from the target domain. This requires the existence of such domain-specific labeled datasets. In many practical scenarios, finding a source domain that is sufficiently related to the target domain can be challenging, and domain adaptation might not yield the desired results due to the vast differences between source and target domains. From this perspective, one-round AL is more flexible as it allocates some budget to label a few samples from the target domain directly, thereby ensuring the model has direct information related to the target and isn't wholly reliant on domain adaptation. We have incorporated the above discussion in Appendix A.
>
> [r1] Chen S, Wang T, Jia R. Zero-Round Active Learning [J]. arXiv preprint arXiv:2107.06703, 2021

---

> > ### Author Response · Authors · 2023-09-02
> >
> > **Q3 [Active learning results on standard data settings]** “*Although noisy and imbalance data settings are more challenge, it is still need to compare the AL methods on standard data settings.*”
> >
> > **A:** We genuinely appreciate the reviewer's input. In response to the suggestion, we have additionally conducted the experiments for the Regular setting on MNIST and CIFAR10, which can be found in Appendix D.5.4 (Table 6). Intriguingly, in this setting, the performance across all methods, including the random selection baseline, are quite close to each other. We conjecture that this is because of the high curation quality of datasets like MNIST and CIFAR10, and we have also seen other studies on multi-round AL with similar observations [1, 2]. Introducing noise and unbalance into the datasets are useful for distinguishing the performance between different AL techniques and such an experiment strategy has also been used in other works [1]. Nevertheless, even in regular data setting, DULO consistently demonstrated a competitive performance compared with other baselines for most of the cases. Moreover, we note that the experiments in Section 5.3.2 are also for the regular settings but targeted datasets with more quality variations. In such contexts, the advantages of DULO become more evident.
> >
> > [1] Killamsetty, Krishnateja, et al. "Glister: Generalization based data subset selection for efficient and robust learning." Proceedings of the AAAI Conference on Artificial Intelligence. Vol. 35. No. 9. 2021.
> >
> > [2] Elenter, Juan, Navid NaderiAlizadeh, and Alejandro Ribeiro. "A lagrangian duality approach to active learning." Advances in Neural Information Processing Systems 35 (2022): 37575-37589.
> >
> > **Q4 [Paired t-test for Table 3 (previously Table 2)]**
> >
> > **A:** We greatly appreciate the reviewer's recommendation. In response, we've incorporated the p-values in the updated version of Table 3. Notably, the majority of these p-values are below the 0.05 threshold, signifying that DULO's improvement over the baselines is statistically significant in most scenarios. The only exception is the IMDb dataset, which appears to present unique challenges for active learning tasks, making it a fertile ground for further exploration and refinement in future studies.

---

### Review · Reviewer_mDNd · 2023-08-14

**Summary Of Contributions:**

- This paper proposes an algorithm for one round Active learning where there is an initial small set of labelled data given and one needs to select datapoint amongst a given set of unlabeled data in one shot.
- They propose a method to learn utility functions for different subsets of the data and then optimize the learned utility function in an approximate way which is a NP hard problem.
- They also formalize the notion of utility transferability across algorithms and propose methods to measure utility transferability for different proxy models.

**Audience:**

Yes

**Claims And Evidence:**

Yes

**Requested Changes:**

- The authors mention there are approaches with multiple rounds of active learning but they might be bad because with few labels, the hypothesizes labels might be wrong. But, it still might be good to add some of them as baselines.
- Also, the authors say that some one round active learning approaches work for linear or kernel regression and the authors also in their work say that deep networks can be approximated by linear methods under appropriate assumptions. So, it might be good to add those as baselines too.
- The authors use a block size for optimization of the utility function and that leads to some approximation to the problem. Is there some way to also quantify how much approximation does that lead to? Probably, for some size that is low enough so that we can try all possible combinations and how how does this blockwise approach hurt?
- I didn’t understand the line on Page 5 in the last paragraph where the authors say we “need to retrain the target for about 5 times the size of the initial labeled data”.
- The authors say that with increasing data set sizes, the points selected do not really matter. But, Theorem 3.1 is for random subsets of data, right? As the dataset becomes larger, I understand that random subsets would have similar accuracy but the best subset still should have a higher accuracy?
- It was hard to follow the formalism in section 4. I am confused about definition 4.1. First, is the transferability defined on algorithms or models? The introduction talked about proxy models. The organization of this section could be improved. It is somewhat hard to read. Moreover, are the proxy models that are experimented with in this work selected by the method proposed in this section?

**Strengths And Weaknesses:**

- The paper is overall quite well written.
- The idea of One Round Active learning seems important and this is one of the few works to study this for general class of models.
- The proposed method is natural and the proposed method outperforms existing baselines on a wide variety of dataset settings like noisy and imbalanced data.

- Some of the writing is not clear and I have detailed specific points below.
- I think some of the baseline comparisons are missing which I have outlined below.

---

> ### Author Response · Authors · 2023-09-02
>
> **Q1 [Additional baselines]** “*The authors mention … add some of them as baselines. Also, the authors say that some one-round active learning approaches work for linear or kernel regression … add those as baselines too.*”
>
> **A:** We sincerely appreciate the constructive feedback from the reviewer. In fact, all of FASS, BADGE, and GLISTER are built upon hypothesized labels. For completeness, we've additionally included CoreSet [1] in our experiments, which do not rely on hypothesized labels. We have also incorporated earlier one-round AL methods for kernel regression models, including Transductive Experiment Design (TED) [2] and Error Bound Minimization (EBM) [3] as additional baselines. We would like to highlight that these methods exhibited considerable computational overhead. Specifically, they did not reach completion within a 24-hour timeframe on CIFAR10 dataset; hence, we present their results solely on the MNIST dataset (that was also the largest dataset these approaches worked in their original papers). This is because these earlier techniques necessitate the computation of the inverse of the Fisher information matrix, a process that is highly computationally heavy for high-dimensional data. We do not compare with the one-round AL technique proposed in [4] as it is specialized to the specific task of image segmentation. We do not compare with the one-round AL technique proposed in [5] as its implementation has not been released, and it is uncertain how the Gram matrix of the infinite NTK is computed based on the paper's description.
>
> In light of the recommendations from Reviewer hRdt, we convert our results representation from line charts to tables. As can be observed from Table 2 and subsequent results, DULO consistently maintains an edge over these additional baselines, underlining its effectiveness.
>
> [1] Sener, Ozan, and Silvio Savarese. "Active learning for convolutional neural networks: A core-set approach." arXiv preprint arXiv:1708.00489 (2017).
>
> [2] Yu, Kai, Jinbo Bi, and Volker Tresp. "Active learning via transductive experimental design." Proceedings of the 23rd international conference on Machine learning. 2006.
>
> [3] Gu, Quanquan, et al. "Selective labeling via error bound minimization." Advances in neural information processing systems (2012).
>
> [4] Jin, Qiuye, et al. "One-shot active learning for image segmentation via contrastive learning and diversity-based sampling." Knowledge-Based Systems 241 (2022): 108278.
>
> [5] Shoham, Neta, and Haim Avron. "Experimental design for overparameterized learning with application to single shot deep active learning." IEEE Transactions on Pattern Analysis and Machine Intelligence (2023).
>
> **Q2 [Error quantification for Blockwise Stochastic Greedy (BSG)?]** *“... quantify how much approximation does that lead to?”*
>
> **A:** We appreciate the reviewer's thoughtful comments. Theoretical error quantification introduced by the Blockwise Stochastic Greedy (BSG) method poses significant challenges. This complexity stems from two primary sources: (1) The intricate higher-order interactions among data points influencing the final model performance, and (2) The discrepancies between the prediction from the data utility model and the actual data utility. Theoretical analysis of the two error sources necessitates examining the interactions between training data and the trained neural networks, which is known to be a highly challenging task. Nevertheless, in Section 5.3.5, we conducted an empirical assessment of how varying block sizes influence active learning performance. Note that when the block size corresponds to the unlabeled pool's size, BSG reduces to the original stochastic greedy algorithm. Our results, showcased in Figure 8, affirm that overly small or large block sizes can compromise the selected data's utility. Such diminishing returns arise because (1) small block sizes may not capture interactions adequately between data points from different blocks, and (2) excessively large block sizes can amplify prediction inaccuracies of the learned data utility model. Moreover, we observed that there is an optimal range of block sizes that ensure **both computational efficiency and high data utility**, and this range approximately equals the size of the initial labeled set. This has informed our heuristic choice for $B$, aiding in striking a balance between computational considerations and the quality of selected subsets.
>
> We have added a pointer in Section 3.1 referring to our ablation study in Section 5.3.5.
>
> **Q3** *“I didn’t understand the line on Page 5 in the last paragraph where the authors say we “need to retrain the target for about 5 times the size of the initial labeled data”*
>
> **A:** We apologize for the confusing sentence. It means the number of utility samples required (i.e., the number of target model retraining required) is of the order of the size of the initial labeled data. We have modified the corresponding texts in the paper.

---

> > ### Author Response · Authors · 2023-09-02
> >
> > **Q4 [Implication of Theorem 3.1 for Active Learning]** *“The authors say that … a higher accuracy?”*
> >
> > **A:** You're right, and we recognize that our original phrasing might not have conveyed our intention. We've taken your feedback into account and updated the manuscript for clarity. Theorem 3.1 suggests that as the dataset grows, the variation in utility scores for randomly selected subsets may diminish, making it challenging for the data utility model to discern between optimal and suboptimal subsets or data points. Hence, focusing on relatively small initial labeled sets is beneficial both in terms of computational efficiency and the effectiveness of data selection (this has also been empirically observed in Section 5.3.5). We genuinely appreciate your keen observation and feedback.
> >
> > **Q5 [Organization of Section 4]** *“It was hard to follow the formalism in section 4. I am confused about definition 4.1. First, is the transferability defined on algorithms or models? The introduction talked about proxy models. The organization of this section could be improved. It is somewhat hard to read. Moreover, are the proxy models that are experimented with in this work selected by the method proposed in this section?”*
> >
> > **A:**
> > For *“is the transferability defined on algorithms or models?”*: The transferability is defined on algorithms. We have modified the mathematical notations in Definition 4.1 for additional clarity. In the community of deep learning, “models” can be a little bit ambiguous: sometimes it refers to the specific instances trained on the data points, and sometimes it refers to the broader notion of the learning algorithm or architecture. We used the term "proxy models" for simplicity and to align with prior works [1]. We appreciate the keen observation and we have added a footnote in the Introduction for additional clarity.
> >
> > For *“The organization of this section could be improved.”*: we have significantly revised and improved the presentation of this section. Specifically, we have added the subtitles and divided subsections for Section 4, and we have moved the experiment in Section 4 to the evaluation section (now in Section 5.2)
> >
> > For *“are the proxy models that are experimented with in this work selected by the method proposed in this section?”*: indeed, the proxy models presented in Table 1 were judiciously chosen by weighing both utility transferability with the target model and computational feasibility. For each dataset, we assessed two proxy model architectures that are sufficiently different for a more meaningful comparison. From these, our selection favored the model that offers a superior balance between transferability and computational efficiency.
> >
> > To provide greater clarity, we've updated Table 1 to include the alternative proxy models considered. When both proxy models displayed similar computational runtime or both were comfortably within our computational constraints, we naturally gravitated towards the model exhibiting superior transferability. In cases where one model might have shown marginally better transferability but came at a considerably steeper computational price (for instance, ResNet18 for CIFAR10), we opted for the model that ensured a more practical runtime.
> >
> > To emphasize the advantage of selecting proxy models with higher utility transferability, we conducted an additional experiment in Appendix D.5.3 where we compare DULO's performance when implemented with our chosen proxy model versus the alternative proxy model (in Table 5). The results clearly show that DULO, when paired with the selected proxy model, outperforms its counterpart with the alternative model. Notably, even when using the alternative proxy model, DULO's performance remains competitive with other baselines.
> > We appreciate your feedback, as it has helped us clarify and enrich the methodology narrative in our paper.
> >
> > [1] Coleman, Cody, et al. "Selection via Proxy: Efficient Data Selection for Deep Learning." International Conference on Learning Representations. 2019.

---

### Review · Reviewer_Diou · 2023-08-18

**Summary Of Contributions:**

In this paper, the authors introduce a new algorithm named DULO tailored for one-round active learning. The key idea is to train a data utility network on labeled training set which serve as a metric for selecting the most useful data points from the unlabeled pool for active learning. Recognizing the computational challenges of the DULO algorithm, the authors propose to use a small proxy model as an alternative. Moreover, the authors discuss how to measure the utility transferability for the selection of proxy models. The authors conduct comprehensive experiments to evaluate the performance of the proposed DULO and provide some ablation studies.

**Audience:**

Yes

**Broader Impact Concerns:**

This paper does not need to add a Broader Impact Statement.

**Claims And Evidence:**

Yes

**Requested Changes:**

- It seems that training T=4000 proxy models to obtain utility samples is impractical especially for large-scale datasets and models. Is it possible to efficiently train for thousands of times while meeting a given computational budget in very large-scale cases?
- The "Measuring Utility Transferability" section explores how to choose a more suitable proxy model to approximate the performance of the target model. However, the experiment section does not provide guidance for selecting a proxy model, but instead specifies it in Table 1. So, where is the role of Measuring Utility Transferability reflected in empirical practice? In other words, how to choose a suitable proxy model when facing a new dataset that can both assess accuracy well and have acceptable running time?
- Does the proxy model also have a trade-off between accuracy and running time, and how to control the risk of overfitting to the small-size validation set?
- It's not very clear why the blockwise stochastic greedy algorithm (BSG) in DULO works, and it seems more like an empirical method. Does the algorithm converge? What is its algorithmic complexity?
- The randomness in BSG (including the selection of blocks B and the selection of subset Z), does it cause instability in the experimental results? To be specific, this paper mentions that the candidate data points to be selected in each iteration of the stochastic greedy algorithm are a smaller, randomly selected subset instead of all unselected data points. Does this smaller, randomly selected subset on the greedy algorithm introduce instability in convergence while accelerating the process?
- Analyze the time complexity of the DULO algorithm. Compare and discuss it with the algorithmic complexity of the baseline methods.
- Besides accuracy, can the experimental section evaluate the running time of baseline methods under the same settings? (Compared to the DULO algorithm's time consumption in Table 1)
- In the 5.2.1 experiment section, do more complex datasets (such as cifar100, IMDb) also show the same tendency?


**Strengths And Weaknesses:**

Strengths:
1. The DULO algorithm introduces a new approach by utilizing data utility prediction to evaluate the scores of subsets, subsequently selecting the optimal subset from the unlabeled pool for active learning.
2. The authors present some strategies, including the proxy model technique and the blockwise stochastic greedy algorithm, to enhance the efficiency of the DULO algorithm.
3. The authors address some of the concerns related to the computational efficiency of the DULO algorithm in Sections 3.2 and 4.

Weakness：
1. The data utility model in the DULO algorithm is trained on T utility sample pairs which are obtained by training T network classifiers separately. Although the authors claim that it's efficiently trainable for thousands of times ($T=4000$ is suggested in the paper), this assumption seems rather impractical, especially in large-scale scenarios.
2. While the paper suggests substituting the target model with a proxy model as a solution to lower down the computational complexity, it doesn't completely solve the computation cost concerns related to the size of utility samples. Moreover, it seems unclear how to select the proxy model empirically (the authors specify proxy models in the experiment section), even if the authors introduce a section named "Measuring Utility Transferability" discussing about how to measure the effectiveness of proxy models.
3. This paper needs to discuss and compare with existing algorithms in terms of time complexity from an algorithmic perspective, and in terms of the tradeoff between accuracy and running time from an experimental perspective.

---

> ### Author Response · Authors · 2023-09-02
>
> **Q1. [Scalability]**
>
> **A:** In this paper, we use a smaller proxy model as a surrogate for the original large target models to facilitate efficient re-training. Additionally, recent advances in fast neural network training (e.g., FFCV [1]) allow one to train a wealth of models on different subsets very efficiently by removing the data loading bottleneck. This technique is used in [2] to improve scalability. The results in our paper do not incorporate such advanced techniques, and we expect that doing so can further improve data utility learning’s efficiency. Another possible way of improving the scalability of large models are **localized retraining** [3] where we only retrain part of the neural networks given the knowledge in large models is often localized. This work lays the groundwork for future research that might consider more sophisticated techniques for scalable real-world deployment.
>
> [1] https://github.com/libffcv/ffcv
>
> [2] Ilyas, Andrew, et al. "Datamodels: Predicting Predictions from Training Data." ICML 2022.
>
> [3] Meng, Kevin, et al. "Locating and editing factual associations in GPT." NeurIPS 2022.
>
> **Q2. [Proxy model selection]** *“How to choose a suitable proxy model when facing a new dataset that can both assess accuracy well and have acceptable running time?”*
>
> **A:** The proxy models presented in Table 1 were judiciously chosen by weighing both utility transferability with the target model and computational feasibility. For each dataset, we assessed two proxy model architectures that are sufficiently different for a more meaningful comparison. From these, our selection favored the model that offers a superior balance between transferability and computational efficiency.
>
> To provide greater clarity, we've updated Table 1 to include the alternative proxy models considered. When both proxy models displayed similar computational runtime or both were comfortably within our computational constraints, we naturally gravitated towards the model exhibiting superior transferability. In cases where one model might have shown marginally better transferability but came at a considerably steeper computational price (for instance, ResNet18 for CIFAR10), we opted for the model that ensured a more practical runtime. We have included an additional paragraph in Section 5.3 to make this selection procedure and considerations clear.
>
> To emphasize the advantage of selecting proxy models with higher utility transferability, we conducted an additional experiment in Appendix D.5.3 where we compare DULO's performance when implemented with our chosen proxy model versus the alternative proxy model (Table 5). The results clearly show that DULO, when paired with the selected proxy model, outperforms its counterpart with the alternative model. Notably, even when using the alternative proxy model, DULO's performance remains competitive with other baselines.
>
> We're grateful for the insightful questions.
>
> **Q3.** *“Does the proxy model also have a trade-off between accuracy and running time?”*
>
> **A:** We appreciate the insightful comment regarding the trade-off between accuracy and running time for the proxy model. Indeed, smaller proxy models may have lower accuracy compared with larger proxy models. However, we would like to stress that the proxy model technique is grounded in the principle of **utility transferability**. As evidenced in past works, the utility of specific data points for one learning algorithm tends to be indicative of their utility for another [1]. In other words, even if the proxy model isn't as accurate as the target model or other larger proxy models, its ability to rank the utility of different data subsets can be strongly correlated with the rankings produced by larger models.
>
> Therefore, to directly address the reviewer's query: while there's an intrinsic trade-off between accuracy and running time for proxy models, for our purposes, the accuracy of the proxy model is secondary. The pivotal factor is its **transferability**. As long as the proxy model can effectively rank the relative utility of different data subsets (which, in our observations, it does), it serves its purpose efficiently. This methodology of prioritizing transferability over sheer accuracy enables us to maintain the quality of active learning decisions while substantially reducing computational demands. Our experience is that there’s no stronger correlation between proxy models’ accuracy and their transferability to the original target models.
>
> [1] Coleman, Cody, et al. "Selection via Proxy: Efficient Data Selection for Deep Learning." International Conference on Learning Representations. 2019.

---

> > ### Author Response · Authors · 2023-09-02
> >
> > **Q4** *“how to control the risk of overfitting to the small-size validation set?”*
> >
> > **A:** Thank you for highlighting the concern of overfitting to small validation sets. This challenge is actually pervasive across many active learning (AL) methods (e.g., [1, 2, 3]), not just our approach. One common way to mitigate the risk is to introduce regularization in the data-selection objective. That is, we can combine our primary AL objective with a regularization term $R(S)$, and we aim to find the subset $S = \text{argmax}_S \widehat U(S) + R(S)$. A simple choice of $R(S)$ is simply a random Gaussian noise; it can serve as a perturbation and ensure that the selection isn't overly optimized for the validation set, thus reducing overfitting risks. We can further improve the robustness of AL performance by leveraging data augmentation. Transforming the data in various ways (e.g., rotations, translations, and other perturbations) can artificially expand the effective size of our validation set, making the model less prone to overfitting the original validation data.
> >
> > [1] Killamsetty, Krishnateja, et al. "Glister: Generalization based data subset selection for efficient and robust learning." Proceedings of the AAAI Conference on Artificial Intelligence. Vol. 35. No. 9. 2021.
> >
> > [2] Yu, Kai, Jinbo Bi, and Volker Tresp. "Active learning via transductive experimental design." Proceedings of the 23rd international conference on Machine learning. 2006.
> >
> > [3] Gu, Quanquan, et al. "Selective labeling via error bound minimization." Advances in neural information processing systems (2012).
> >
> >
> > **Q5 [Why does stochastic greedy algorithm (BSG) work? Does it converge? What is its algorithmic complexity?]**
> >
> > **A:**
> >
> > **(1) Convergence of the algorithm?** We'd like to clarify that in the context of (stochastic) greedy algorithms for set function maximization, the concept of "convergence" is not typically applicable. Such algorithms are designed to terminate upon collecting a subset of a specified target size – in our case, this size corresponds to the labeling budget for active learning.
> >
> > **(2) Computational Complexity:** As noted in Section 3.1, the computational complexity of the BSG algorithm is $O(|U|)$ in terms of the number of evaluations of $\widehat U$ (and more specifically, it’s $O(|U| \log(1/\epsilon))$ where $\epsilon$ is a hyperparameter in Algorithm 1). However, it is important to recognize that the runtime for the evaluation of $\widehat U(S)$ grows substantially with the size of $S$. When $S$ becomes excessively large, not only does it increase the computational burden, but it can also amplify the prediction inaccuracies of the learned data utility model.
> >
> > **(3) Rationale for Blockwise Selection:** This complexity challenge led us to adopt the blockwise approach. With BSG, we can balance between computational efficiency and the quality of data utility selection. In essence, by choosing a block size, we ensure that we're neither evaluating excessively large sets (which would be computationally challenging and could lead to inaccurate utility estimations) nor overly small sets (which might miss out on capturing data interactions).
> >
> > **(4) Empirical Ablation Study:** We endeavored to provide empirical evidence to validate the effectiveness of BSG. In Section 5.3.5, we dedicated an empirical assessment to understand the influence of varying block sizes on AL performance. Our findings, as depicted in Figure 8 show that there is an optimal range for block sizes, ensuring a balance between computational efficiency and the utility of selected data. The optimal range for block sizes is close to the size of the initial labeled set. This has informed our heuristic choice for $B$, aiding in striking a balance between computational considerations and the quality of selected subsets.
> >
> > In conclusion, we appreciate the feedback and acknowledge that, like many algorithms in machine learning, the BSG method in DULO does rely on empirical validations. However, our findings make a strong case for its efficacy in the context of active learning.

---

> > > ### Author Response · Authors · 2023-09-02
> > >
> > > **Q6 [Randomness in BSG algorithm?]** *“The randomness in BSG … while accelerating the process?”*
> > >
> > > **A:**
> > > As we mentioned in the earlier question, in the context of greedy algorithms for set function maximization, the conventional notion of "convergence" is not applicable. Greedy algorithms iteratively select elements to maximize a given objective function over a specified set. They terminate once the desired subset size (in our context, the labeling budget) is reached, rather than converging to a particular value.
> > >
> > > Indeed, like most of the AL techniques, the data selection process of DULO is also randomized. To account for the variability introduced by this randomness, all our experimental outcomes include error bars. While there's an element of randomness in the process, our empirical results consistently demonstrate the effectiveness and robustness of our approach across multiple runs. Furthermore, the inherent randomness can be beneficial as it can mitigate overfitting to the validation set. By not deterministically selecting the same subset in each run, the algorithm is less likely to consistently over-weigh or under-weigh specific instances, making it more robust to potential biases or anomalies in the validation set.
> > >
> > > **Q7 [Time complexity of DULO, and comparison with the algorithmic complexity of baseline algorithms]**
> > >
> > > **A:**
> > > **DULO's Runtime**
> > > - **Learning Stage**: The runtime in this stage is primarily determined by the number of utility samples being collected. It's crucial to note that providing an analytical derivation of the sample complexity required for the desired accuracy of the data utility model is a profound challenge. This complexity arises primarily due to the intricacies associated with determining the sample complexity of neural networks—an issue well-documented in the literature as particularly challenging. Such challenges aren't exclusive to DULO; they span the active learning literature in the context of deep learning. For instance, GLISTER bases its computational complexity on a predetermined number of epochs, rather than on a "target error rate for bi-level optimization”.
> > > - **Optimization Stage**: As we've mentioned in earlier questions, the computational complexity of the BSG algorithm for DULO stands at $O(|U|)$ in terms of the number of evaluations of $\widehat U$.
> > >
> > > **Comparing Algorithmic Complexity Across AL Methods.** Direct comparisons of algorithmic complexities among various AL techniques may not yield meaningful insights. As previously highlighted, much of the AL literature tends to anchor its complexity analysis to pre-defined quantities such as "the number of epochs in training". Ideally, to make a meaningful comparison of time complexities between two algorithms, one would measure them against a "target error rate." However, such an analysis is almost impossible for active learning approaches tailored for deep learning due to the intricate nature of neural networks. Additionally, these complexities are often framed in terms of distinct basic operations. For instance, while our method delineates complexity via the lens of utility sample collections and evaluations of $\widehat U$, GLISTER [1] states its complexity through the number of gradient updates in bi-level optimization.
> > >
> > > [1] Killamsetty, Krishnateja, et al. "Glister: Generalization based data subset selection for efficient and robust learning." Proceedings of the AAAI Conference on Artificial Intelligence. Vol. 35. No. 9. 2021.

---

> > > > ### Author Response · Authors · 2023-09-02
> > > >
> > > > **Q8 [Runtime of baseline algorithms?]**
> > > >
> > > > **A:** We sincerely appreciate the reviewer's insightful suggestion. To address this, we have included the running clock time for all baselines we use in Appendix D.5.5 (in Table 7). While DULO is not as efficient as multi-round AL techniques, it aligns with our expectations for a one-round AL approach. Given the intrinsic nature of one-round AL, it demands more computations and greater information extraction from the initial labeled dataset. However, it's worth highlighting that DULO holds a marked computational edge over traditional one-round AL methods for linear models (we included the experiment for these specialized one-round AL techniques during the rebuttal period due to the request of Reviewer mDNd and hRdt). This is because these earlier techniques necessitate the computation of the inverse of the Fisher information matrix, a process that is highly computationally heavy for high-dimensional data.
> > > >
> > > > Furthermore, it is crucial to emphasize that in practical AL scenarios, computational time might not always be the primary concern. Instead, the time spent interacting with domain experts for labeling often dominates the runtime overhead. Therefore, while DULO may not be as efficient as those adapted from multi-round AL techniques, its outstanding performance makes it an excellent choice in one-round AL settings.
> > > >
> > > >
> > > > **Q9** *“In the 5.2.1 experiment section (now at Section 5.1), do more complex datasets (such as cifar100, IMDb) also show the same tendency?”*
> > > >
> > > > **A:** We have included additional plots for CIFAR100 and IMDb dataset in Appendix D.2 in Figure 9. As we can see, these more intricate datasets align with previously observed tendencies: $\widehat U$ tends to slightly underestimate the utility for high-quality subsets and overestimate for low-quality subsets. Notwithstanding, the overarching trend remains consistent, with the Spearman correlation coefficient between the predicted and actual model performance is $0.828$ for CIFAR100 and $0.86$ for IMDb. We conjecture that due to the inherent challenges of these tasks, neural networks might tend to fit a constant value approximating the mean of the target values. Delving deeper into the intricacies of this phenomenon offers an intriguing avenue for future research.

---

### Author Response · Authors · 2023-09-02
**Summary of paper changes**

Dear Reviewers,


We thank all reviewers for providing helpful feedback and suggestions. We considered the reviews carefully and modified our paper accordingly. All modifications are highlighted in red. Here’s a summary of our major revision to the paper:
- **Organizations**: We improved the organization of Section 4, where we moved the experiments for comparing VMC and LBEB heuristics to Section 5, and we polished the overview paragraph and added subsections and paragraph titles.
- **Additional baselines**: We have included CoreSet [1] as well as the earlier one-round AL techniques for linear models (where we evaluate the performance of neural networks trained on the selected data points) [2, 3].
- **Data Utility Prediction for more complex datasets**: We additionally include the utility value prediction plot for CIFAR100 and IMDb datasets in Appendix D.2 (Figure 9).
- **Active learning results on clean MNIST and CIFAR10 dataset**: We additionally include the one-round active learning results on the clean MNIST and CIFAR10 datasets in Appendix D.5.4 (Table 6).
- **Use table instead of figures to represent experiment results**: We have changed the original figures for active learning’s results into table representations for clarity.
- **Selection of proxy models**: We have included a paragraph in Section 5.3 titled *“Choosing proxy models”* which explains our approach for choosing the proxy models based on the tradeoff between the computational time and transferability. We've updated Table 1 to include the alternative proxy models considered. To further emphasize the advantage of selecting proxy models with higher utility transferability, in Appendix D.5.3 (Table 5) we compare DULO's performance when implemented with our chosen proxy model versus the alternative proxy model.
- **Running clock time of baseline algorithms**: We have included the clock time of baseline algorithms in Appendix D.5.5 (Table 7).

[1] Sener, Ozan, and Silvio Savarese. "Active learning for convolutional neural networks: A core-set approach." arXiv preprint arXiv:1708.00489 (2017).

[2] Yu, Kai, Jinbo Bi, and Volker Tresp. "Active learning via transductive experimental design." Proceedings of the 23rd international conference on Machine learning. 2006.

[3] Gu, Quanquan, et al. "Selective labeling via error bound minimization." Advances in neural information processing systems (2012).

---

### Decision · Action_Editors · 2023-10-06

**Recommendation:** Accept as is

**Comment:**

The main claims are solid, the paper is well-written, and it has an audience in the TMLR community. The authors have responded in detail to the reviewers' requests, and all of the reviewers were in favor of acceptance following the author response. I agree with them, and feel it is appropriate to accept the paper as-is.

*Dear authors:* Please do not forget to change the red-colored text back to black text for the final version.

**Audience:**

Yes. Active learning is an important area of machine learning with obvious practical applications. The authors' proposed framework and formulation is quite general and open to all sorts of domain-specific specializations.

**Claims And Evidence:**

The main claims in this paper are centered around the proposal and evaluation of a method for doing active learning with just a single query, with access to a presumably small set of labeled data upon which the evaluation of "data utility" is made. It is based on this utility that the query is made, with the help of a computationally congenial "proxy" model. All the basic ideas are intuitive, and while there is certainly conceptual overlap with existing work, the authors are clear about what is distinct here, and their description of the proposed framework is in my opinion quite lucid. They propose a framework for doing one-round active learning, and have empirically found some conditions under which their approach is more effective than traditional techniques; these facts are in line with their main claims, which are thus solid.